# High biodiversity in a benzene-degrading nitrate-reducing culture is sustained by a few primary consumers

Chrats Melkonian [1✉], Lucas Fillinger [2], Siavash Atashgahi[3], Ulisses Nunes da Rocha[4], Esther Kuiper[1], Brett Olivier [1], Martin Braster[1], Willi Gottstein[1], Rick Helmus [5], John R. Parsons[5], Hauke Smidt [3], Marcelle van der Waals[6], Jan Gerritse[6], Bernd W. Brandt [7], Wilfred F. M. Röling[1,8], Douwe Molenaar[1] & Rob J. M. van Spanning [1✉]

A key question in microbial ecology is what the driving forces behind the persistence of large biodiversity in natural environments are. We studied a microbial community with more than 100 different types of species which evolved in a 15-years old bioreactor with benzene as the main carbon and energy source and nitrate as the electron acceptor. Using genome-centric metagenomics plus metatranscriptomics, we demonstrate that most of the community members likely feed on metabolic left-overs or on necromass while only a few of them, from families *Rhodocyclaceae* and *Peptococcaceae*, are candidates to degrade benzene. We verify with an additional succession experiment using metabolomics and metabarcoding that these few community members are the actual drivers of benzene degradation. As such, we hypothesize that high species richness is maintained and the complexity of a natural community is stabilized in a controlled environment by the interdependencies between the few benzene degraders and the rest of the community members, ultimately resulting in a food web with different trophic levels.

[1] Department of Molecular Cell Biology, AIMMS, Vrije Universiteit Amsterdam, Amsterdam, The Netherlands. [2] Department of Functional and Evolutionary Ecology, University of Vienna, Vienna, Austria. [3] Laboratory of Microbiology, Wageningen University & Research, Wageningen, The Netherlands. [4] Department of Environmental Microbiology, Helmholtz Centre for Environmental Research, Leipzig, Germany. [5] Institute for Biodiversity and Ecosystem Dynamics, University of Amsterdam, Amsterdam, The Netherlands. [6] Unit Subsurface and Groundwater Systems, Deltares, Utrecht, The Netherlands. [7] Department of Preventive Dentistry, Academic Centre for Dentistry Amsterdam, University of Amsterdam and Vrije Universiteit Amsterdam, Amsterdam, The Netherlands. [8]Deceased: Wilfred F. M. Röling. ✉email: chrats.melkonian@gmail.com; rob.van.spanning@vu.nl

Microbes typically live in complex and diverse communities[1–3] the richness of which may in part be determined by environmental conditions like the availability of suitable carbon and energy sources and the spatial and temporal variability of this environment[4,5]. Besides environmental conditions, additional ecological factors determine the structure, activity, and succession of these communities, such as competition for resources, cooperation during the exchange of products, and inhibition by chemical warfare[6]. The extent to which environmental effects and interactions with other organisms can determine species richness in a community is still an open question[4,7,8]. Here we study a microbial community living in an anaerobic fixed film bioreactor with benzene as the main source for carbon and energy and nitrate as the terminal electron acceptor, thereby creating a relatively simple environment. This microbial community was enriched from soil samples of a benzene-contaminated industrial site and has been maintained already for 15 years[9], which is enough time to purge a putative initial diversity by mechanisms like competitive exclusion and random fluctuations[10–12]. Nevertheless, the culture is currently remarkably rich in species[13,14]. Spatial and temporal heterogeneity as well as the variety of interactions between organisms are then created and maintained by the community itself, aided by mechanisms like wall adherence and patch formation.

Over the years, a biofilm developed on the interior glass wall of the bioreactor, hosting more than 100 different types of operational taxonomic unit (OTU) based on 16S rRNA sequencing[14]. It has been hypothesized that these types of microbial communities may hold many types of interactions between key players in benzene degradation and other members of the community[15]. A recent study on the same bioreactor revealed high levels of transcripts for an anaerobic benzene carboxylase and a benzoate-coenzyme A ligase produced by *Peptococcaceae*[13]. This finding was in line with other research on benzene degradation where this species was a key player in anaerobic benzene degradation[16,17]. Benzene is an aromatic hydrocarbon, which occurs in crude oil and petroleum products like fuels. Due to its high toxicity and water solubility, benzene is of major concern as an environmental contaminant[18]. Microbes can efficiently open the ring-structure of benzene in the presence of oxygen[19]. However, in hydrocarbon-contaminated subsurface environments, oxygen is rapidly depleted[20,21] leading to anaerobic benzene degradation, which takes place at a lower rate[20,22]. Although the biochemistry of anaerobic benzene degradation is relatively well-known[22–24], the related metabolic pathways are less clear. A comprehensive overview was recently given by Meckenstock et al.[25] and it is also a field of active research[9,15,16,26–32].

The aim of this work was to get a more fundamental understanding of the diversity, structure, metabolic potential and dynamics of the anaerobic benzene-degrading microbial community. More specifically, we aimed at getting insight in the driving forces behind the persistence of large biodiversity in natural environments. We used metagenomics to obtain metagenome-assembled genomes (MAGs), which are assumed to be accurate representations of genomes of individual species[33]. The inferred functions of their genes indicated the potential physiological properties of these species[34]. Metatranscriptomes originating from the biofilm and the liquid phase of the bioreactor were mapped to these MAGs to obtain a measure of their global activity, as well as of the activity of individual genes and pathways. This integrated approach yields a view of the abundances, phenotypes, and activities of these species in these phases[35–37]. An additional experiment was designed to independently identify the main organisms that drive anaerobic benzene degradation as well to explore the metabolism of the culture. To this end, we inoculated a series of batch cultures of the

15 year-old microbial community at low cell densities. The samples were sacrificed over time and analyzed for metabolomics on the one hand, and for their community composition based on 16S rRNA gene amplicon sequencing on the other hand. As such we (i) identified the drivers for benzene degradation, (ii) got insight in the niches of the community members in the bioreactor and (iii) hypothesized about the relevance of niche partitioning and microbial interactions in order to explain the unexpected diversity of species in a bioreactor fed with benzene as main source of carbon and energy.

## Results

**Diversity and activity of the anaerobic benzene degrading community.** We reconstructed 111 MAGs from the metagenomes derived from two samples taken from the biofilm of the culture. From these, 47 high-quality MAGs were selected (Methods "Metagenomics analysis", Supplementary Data 1). The transcriptomes obtained from samples taken from the biofilm and the liquid phase of the culture were mapped to the predicted genes of all 111 MAGs. Both in the biofilm and the liquid samples the abundance of RNA mapped to MAGs correlated positively with the abundance of the DNA assigned to the MAGs (Supplementary Fig. 1A, B, Supplementary Data 1). The specific RNA abundance per MAG was calculated in biofilm and liquid phase samples as a measure of its transcriptional activity (Eqs. (1)–(3)). In Fig. 1a the specific RNA abundance of biofilm versus liquid revealed a gradient of transcriptional activity (high/intermediate/low), with MAGs 3 and 9 displaying the highest activity, a dozen displaying intermediate activity and the majority displaying low activity. The same conclusion can be drawn when transcriptional activity is calculated as the percentage of transcribed genes per MAG. 32 out of the 47 MAGs were found to have a significantly higher specific RNA abundance in samples from the biofilm compared to those from the liquid phase with the exception of four MAGs, including MAG 9, which had the highest overall transcriptional activity in both phases (Supplementary Fig. 2).

**Global functional groups and their relation to taxonomic groups.** The 47 high quality MAGs were further analyzed by annotating predicted genes using orthology relationships. The presence and absence of genes in the Kyoto Encyclopedia of Genes and Genomes (KEGG) Orthology (KO) groups was used to determine potential functional groups of MAGs. Based on these results, the MAGs were divided into 8 clusters, which were classified into three main groups based on a UMAP dimension reduction. In Fig. 1b the potential functional landscape is visualized. MAGs of group A (Clusters 3 and 8) and group B (Clusters 4, 6 and 7) were more similar in terms of identified function compared to group C (Clusters 1, 2 and 5). Furthermore, members of group C were found to have a significantly higher absolute number of annotated KOs, but not a significantly different ratio of KO annotations (Supplementary Fig. 3). Only seven out of the 47 selected MAGs were found to have a high specific RNA abundance and high transcribed KO ratio (Supplementary Fig. 3). Those seven MAGs are distributed over the three groups (Fig. 1b; group A: 9, 1, 5; group B: 18, 20 and 3; group C: 6). We considered these MAGs as putative dominant members of the community and prime candidates to explore their transcription profile in more detail.

The overall taxonomic grouping of the 47 selected MAGs corresponded well with the functional grouping. Functional group A has seven members of the Chloroflexi, four of the Actinobacteria and one of the Firmicutes; group B is composed of 11 members of the Bacteroidetes, three of the Gemmatimonadetes, two of the Verrucomicrobia and one member each of the

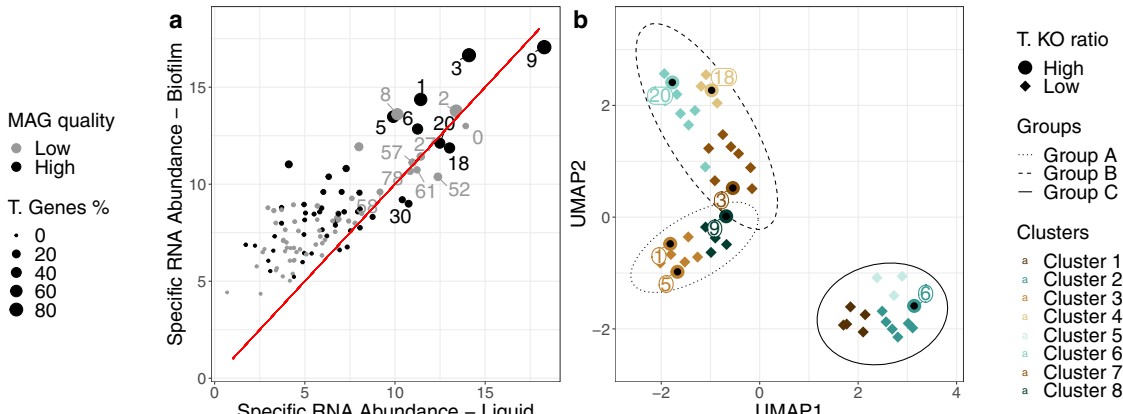

**Fig. 1 Relationship of MAGs in biofilm and liquid according to specific RNA abundance and their functional grouping. a** Relationship between specific RNA abundance in biofilm and liquid for all MAGs. The black-coloured points indicate high-quality MAGs which were used for further analysis. The relative placement of the points with the red diagonal line reveals in which phase the specific RNA abundance is higher. The size reflects the percentage of identified genes with mapped mRNA. Note that MAG 9 occupies the highest specific RNA abundance in both phases followed by MAG 3. MAGs 1, 2, 5, 6, 8, 18 and 20 have a medium level of specific RNA abundance. **b** Functional landscape of the selected MAGs. The colours of the points indicate 8 functional clusters of MAGs, while the ellipses further gather them into three groups. The shape of the points indicates the ratio of the transcribed (T.) KOs (KEGG Orthologs) where the circular points are from the dominant members of the community (as defined in the text) and the rhombus points are from the non-dominant members of the community. The numbers next to the circular points indicate the dominant MAGs. For the dimensional reduction and clustering, we used Uniform Manifold Approximation and Projection (UMAP) and affinity propagation, respectively, on the binary (presence/absence) KO matrix. Note that the dominant MAGs are distributed across the functional groups.

Acidobacteria, of the Armatimonadetes, of the Myxococci and of the Planctomycetes; and group C is composed of 15 members of the Proteobacteria (Supplementary Fig. 4 and Supplementary Table 1). An additional 22 MAGs showed significant matches with the Silva 132 database of ribosomal RNA (Supplementary Table 1). Further details about the taxonomy of the MAGs is presented in the Supplementary Sections 1.10 and 2.2.

**Genomic potential of the community members**. A substantial part of KOs is shared between the three functional groups (2807 out of 6449 unique KOs), while many other KOs are unique for each group. Moreover, the different clusters within each group showed differences in their KO profiles (Supplementary Fig. 5). Feature selection revealed 193 KOs, which further mapped into 16 KEGG pathways as discriminators for the three functional groups (Supplementary Fig. 6A, B). We performed a number of targeted searches on potential functionality (genes) and activity (mRNAs) of each of the MAGs. We found that MAGs 3, 6 and 9, as well as those from the Proteobacteria have more and relatively highly expressed genes required for motility and/or adhesion, such as the biosynthesis of flagella and/or pilus systems (Supplementary Figs. 7 and 8). Notably, all MAGs from the Chloroflexi expressed relatively high levels of mRNAs encoding peptidases, extracellular solute-binding proteins and specific ABC-type transporters. This property is shared with one of the Bacteroidetes (MAG 18). Members of cluster 2 from the Proteobacteria were found to contain the majority of genes encoding secretion systems, with MAG 36 (*Hydrogenophagus*) having genes for even different types of these systems (Supplementary Fig. 9). Additionally, we noted relatively high concentrations of mRNAs encoding a type II secretion system from MAG 66 (Actinobacteria), types II and III from MAG 22 (Armatimonadetes) and type VI from MAG 7 (Acidobacteria) and MAG 10 (Ignavibacteria). Genes encoding nitric oxide dismutase (NOD) were found in MAGs 33 (re-assigned on MAG 0), 34 and 71. From those, only MAG 34 passed the quality control and was classified as an unknown member of the $\gamma$-proteobacteria (Supplementary Fig. 34). MAG 3 belongs to a member of the Planctomycetes with the highest similarity to *Candidatus* Kuenenia stuttgartiensis

(average nucleotide identity 94.7%). The MAG has the key genes for anaerobic ammonium oxidation (anammox), hydrazine synthase and dehydrogenase. A further description of the metabolic and structural characteristics of the MAGs is available in Supplementary Section 2.3.

**Metabolism of benzene**. In order to characterize the metabolic potential of MAGs, we selected all known pathways involved in anaerobic benzene degradation, central carbon metabolism and nitrate reduction (Supplementary Section 1.14 for details, Supplementary Figs. 12–24). A network of these pathways using lumped reactions was created for visualization (Supplementary Figs. 25–29). Using these custom pathways and corresponding networks, we found only two MAGs from the dominant species with the potential to activate and further degrade benzene anaerobically (primary consumers, Fig. 2). These are MAGs 6 and 9 that were identified as members of *Rhodocyclaceae* and *Peptococcaceae* and belong to groups C and A, respectively (Fig. 2c). In both MAGs we observed a high expression of genes involved in anaerobic benzene degradation, such as UbiD/UbiX-related carboxylases, benzoate-CoA ligase and benzoyl-CoA reductase. Some non-dominant Proteobacteria (MAGs 19, 25, 47 and 68) have the potential to be primary consumers as well (Fig. 2d). Surprisingly, MAGs 19, 36, 47, 56 and 68 showed high expression of genes involved in the aerobic degradation of aromatic compounds, including benzoyl-CoA oxygenase. Moreover, MAGs 19 and 68 showed high expression of genes encoding protocatechuate 4,5-dioxygenase, another oxygen demanding enzyme central in 3,4-dihydroxybenzoate metabolism. For further details on metabolism of benzene we refer to Supplementary Section 2.5.

**Nitrogen cycling**. Nitrate is the main electron acceptor supplied to the benzene-degrading microbial community. The major processes of respiratory electron flow to nitrate and nitrite are (i) dissimilatory reduction of nitrate or nitrite to ammonium (DNRA) and (ii) sequential reduction of nitrate to dinitrogen gas (denitrification). Most of the 47 MAGs have the potential to perform DNRA (8 MAGs), denitrification (31 MAGs), or both (5 MAGs). Only 3 MAGs, 3, 7 and 36, are unable to do so although

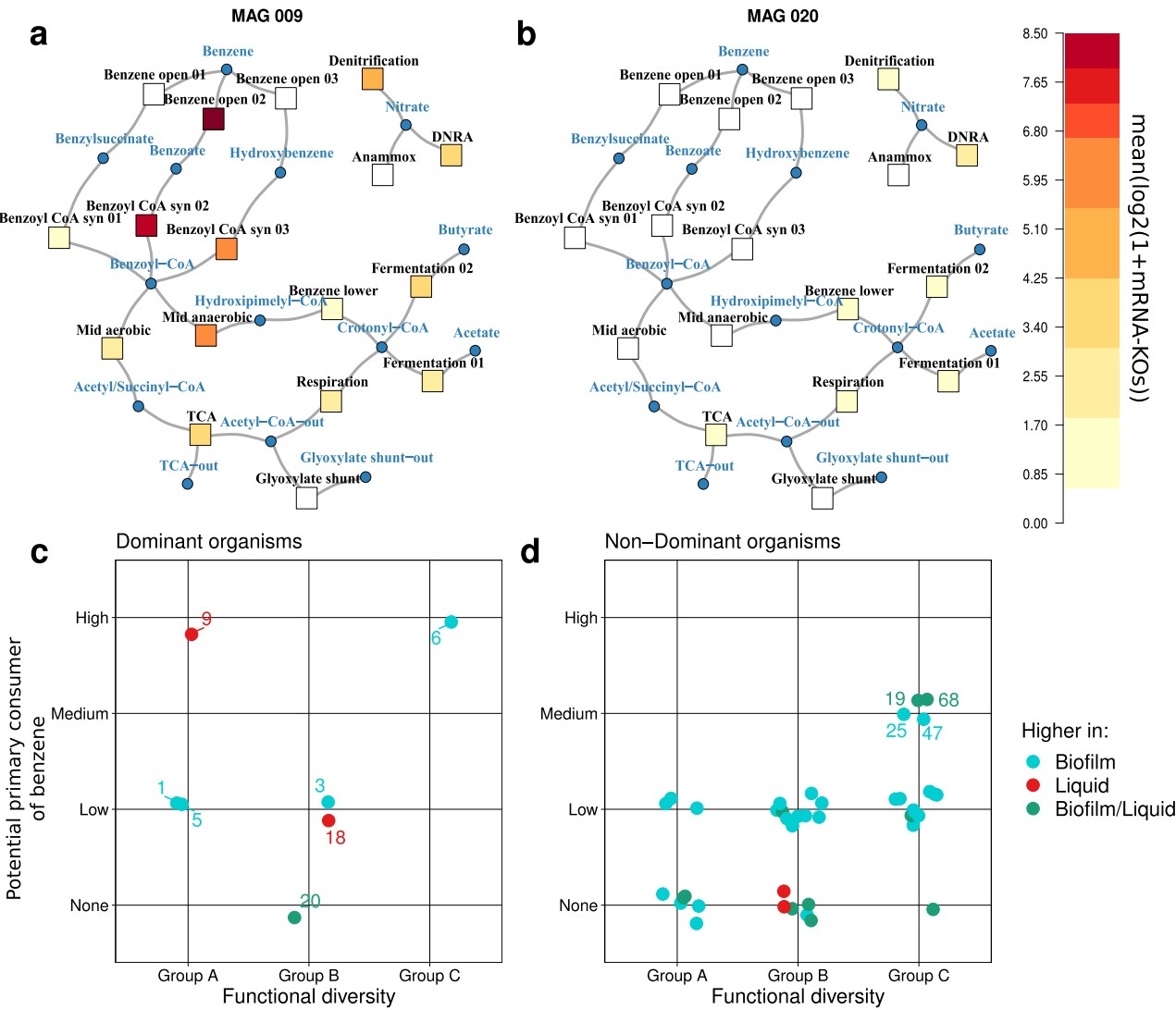

**Fig. 2 Metabolic heatmaps of anaerobic benzene degradation in MAGs 9 (*Peptococcaceae*), and 20 (*Ignavibacteriaceae*) and relationship between functional diversity and potential to degrade benzene. a** Heatmaps of MAGs 9 and **b** 20, which are dominant members of the community with the highest and lowest potential for benzene metabolism, respectively. The square nodes represent the average transcription of a lumped group of genes that encode enzymes of each branch, and the circular nodes correspond to intermediate compounds. The two lower figures are overviews illustrating the potential of MAGs to be a primary consumer of benzene (the ability to activate and further degrade benzene) based on a custom classification for **c** dominant and **d** non-dominant MAGs. The colours indicate the culture phase in which MAGs show significantly higher specific RNA abundance. The functional diversity is derived from the three groups of Fig. 1. Note that only a few MAGs are assigned as potential primary consumers of benzene, while the majority of the MAGs are not, including dominant MAGs such as 1, 3, 5, 18 and 20. Data are shown jittered for both axes.

they have a gene encoding a nitrate reductase but lack one encoding a nitrite reductase (Supplementary Section 2.1 for details).

**Succession of communities in batch grown cultures**. A succession experiment was performed to investigate the metabolism of the community as well as to identify the drivers of benzene degradation after giving them a fresh start. For that, we grew highly diluted batch cultures from the original bioreactor with benzene as carbon and energy source and nitrate as electron acceptor. The cultures were sacrificed at different time intervals up to 34 days after inoculation and analyzed for the community composition on the one hand and concentration of the metabolites on the other hand. Over this time period, individual batch cultures displayed high variability in rates of benzene consumption (Fig. 3a). Some cultures depleted benzene within 22 to

26 days, whereas others had not even started consuming benzene after 34 days. We identified three stages based on the pattern of nitrate consumption (Fig. 3a). Stage 1 was characterized by a lack of benzene consumption. In stage 2, benzene was consumed until a residual concentration of approximately 0.04 mM, and in stage 3 benzene was depleted down to a residual concentration of around 0.005 mM. Notably, the differences in benzene degradation rates could not be explained by differences in cell densities. As shown in Fig. 3c, cell densities increased quickly after inoculation and then slowly rose up to 20 days. After that time, there were cultures in which benzene had been depleted, but with a range of densities between $10^5$ and $2 \times 10^6$ cells per mL. We also observed cultures with high cell density in which benzene degradation did not occur. It may well be that these cultures consumed the acids and vitamins as alternative carbon and energy source (Fig. 3d). Figure 3c shows how nitrate consumption, nitrite production and benzene consumption are linked. The

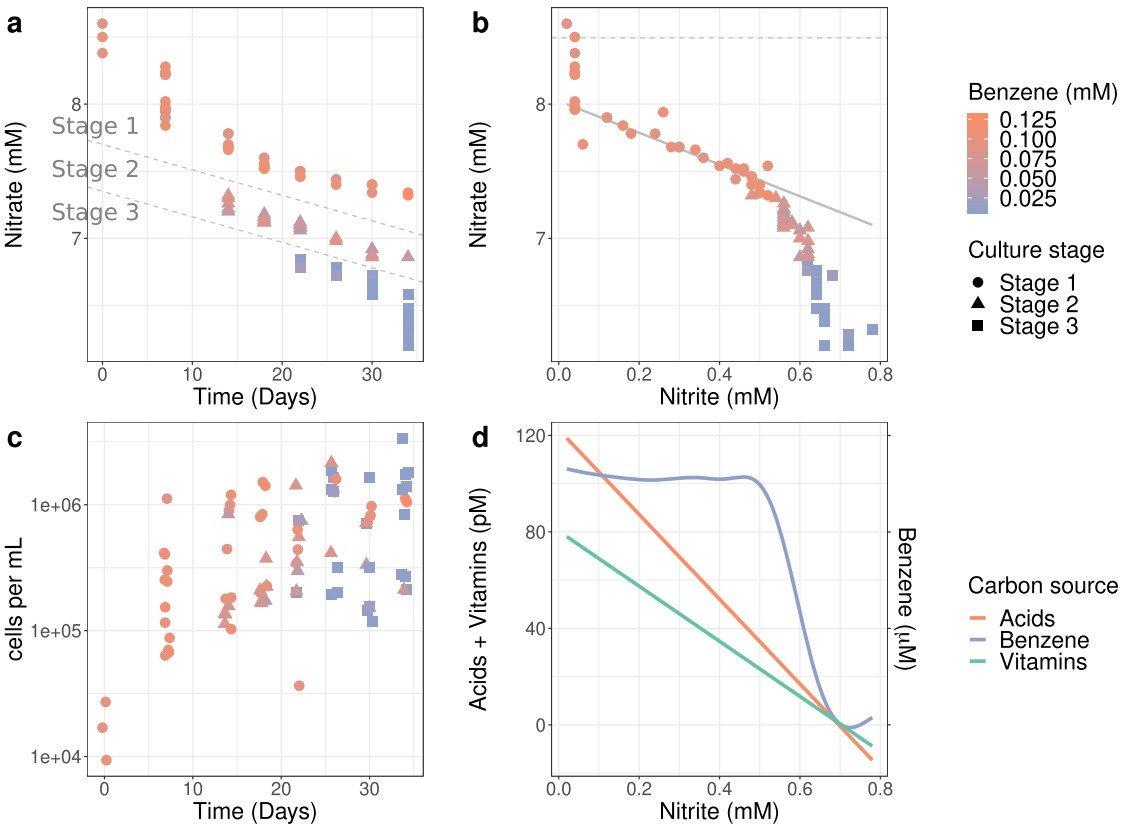

**Fig. 3 Progression of benzene degradation in time during the succession experiment.** Data analyses of cultures sampled at different time points after inoculation. **a** Decrease of the nitrate and benzene concentrations. Stages are separated by the grey lines and explained in the text. **b** Relationships between the onset of benzene degradation, nitrate and nitrite levels and culture stage. The dashed line indicates the average of the measured starting concentration of nitrate. The sloped grey line indicates a stoichiometrical (1:1) ratio of consumption and production of nitrate and nitrite. The intercept of this line was chosen by eye. Note that the culture stage and benzene concentration change only above a nitrite concentration of 0.5 mM or a nitrate concentration below 7.3 mM. **c** Correlations of cell density, stage and benzene concentration. Data points are jittered relative to the time axis. **d** Relation between acids (nicotinic, pantothenic and para-aminobenzoic acids), vitamins (biotin and vitamin B12) and benzene consumption with accompanying nitrite levels (Supplementary Fig. 30 for detailed view on each carbon source). The functions were fitted with a generalized additive model with integrated smoothness estimation. Acids and vitamins were additives in the original media. Numbers and increments of left and right y-axis are the same. Note the initial consumption of acids followed by vitamins, only after which benzene consumption starts. The sudden drop in the fitted benzene consumption plot coincides with the decrease in nitrite production in stage 2 (Supplementary Fig. 31 for details).

data points of the individual cultures form a continuum, suggesting a single deterministic process of benzene degradation. The three stages corresponded to segments of this continuum. Cultures in the early phase of stage 1 consume nitrate without accumulation of nitrite. In a later phase of stage 1, the cultures consumed nitrate and produced nitrite at a stoichiometric 1:1 ratio until a nitrate concentration of about 0.5 mM, indicated by the sloped grey line in Fig. 3c. The later phase corresponds to stages 2 and 3, where nitrate was further consumed whereas the nitrite concentration increased at a lower rate up to 0.7 mM and 0.8 mM, respectively (Supplementary Fig. 31). Benzene is consumed only in this phase. We measured concentrations for some of the vitamins in the medium samples (i.e. those listed in Supplementary Data 2 and Supplementary Table 2). However, we could not detect any of the intermediates of benzene metabolism (Supplementary Tables 3 and 4) in the medium samples.

**Community composition of the batch grown cultures**. We determined the community composition in each of the cultures by amplicon sequencing of the 16S rRNA gene. Reads corresponding to 192 OTUs were identified and, after correction for 16S rRNA gene copy numbers, expressed as fractions of total corrected reads. Using the measured total cell densities, OTU-

specific cell densities were calculated for each sample. We subsequently identified a transition in community composition as well in the OTUs of which the cell densities correlated with the three stages of culture development (Fig. 4 and Supplementary Fig. 32). The strongest correlations were observed for OTU624837510 and OTU91680185 whose sequences belong to the genus *Thermincola* and the family of the *Peptococcaceae*, respectively (Table 1 and Supplementary Table 5). Not surprisingly, the representative sequence of OTU624837510 is found to be identical to the 16S rRNA gene of MAG 9 from the original bioreactor. Additionally, OTU717462002 correlated well with the progression of stage 2 to 3. It was identified as a member of the *Rhodocyclaceae* and taxonomically identical to MAG 6. We further found significant matches between the 16S sequences of another seven of the selected MAGs and of the apparently corresponding OTUs from the succession experiment (all 100 % identity, Supplementary Table 1).

**Discussion**

Here we applied a genome-centric metagenomics combined with metatranscriptomics approach to obtained insights in the community diversity, structure, function and dynamics of an anaerobic microbial community that developed in a bioreactor during

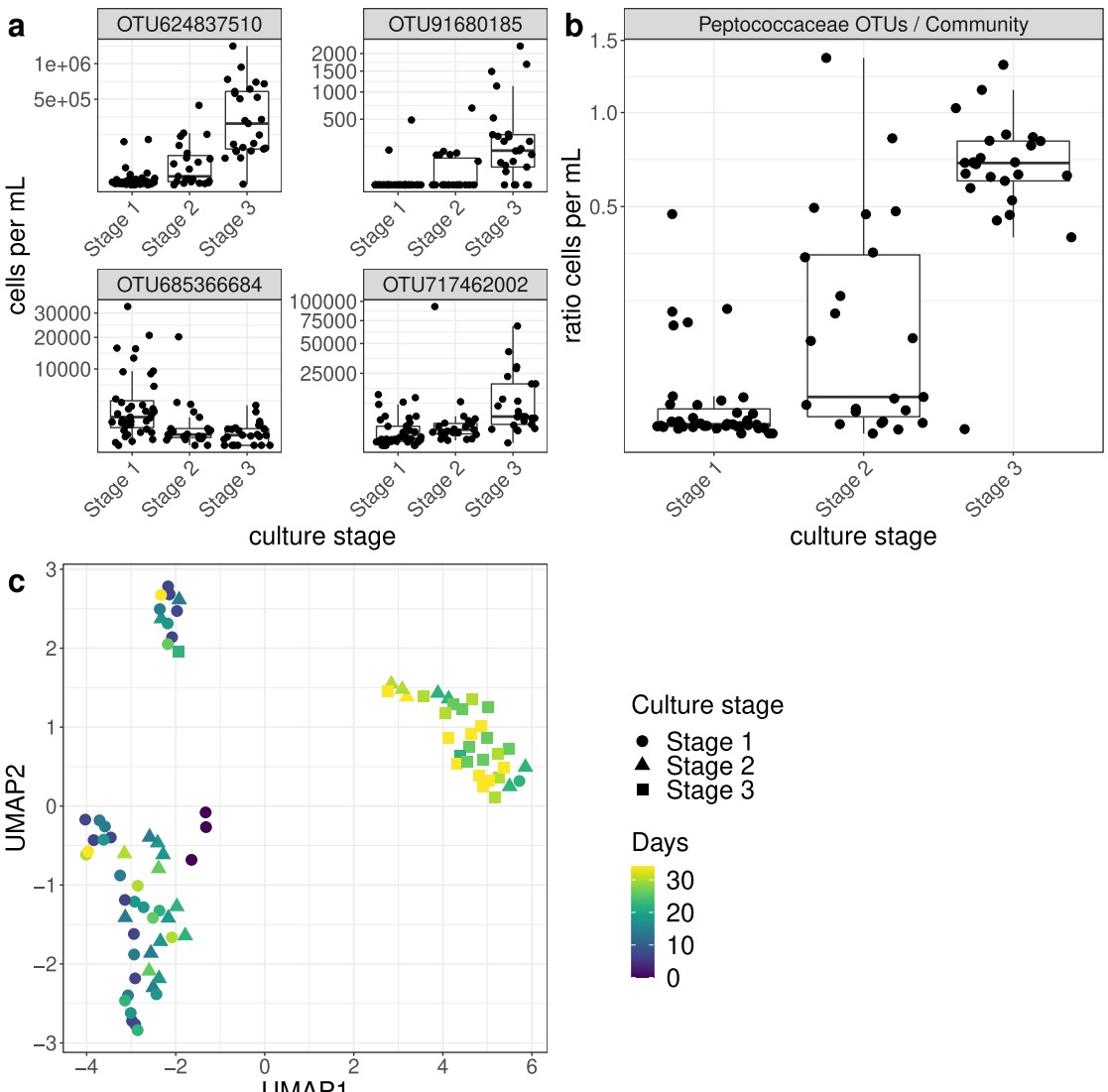

**Fig. 4 *Peptococcaceae* drive changes of the community composition and the progression of benzene degradation. a** Relationship between culture stage and cell abundance of 4 selected OTUs (by random forest variable importance and PERMANOVA analysis). **b** Relationship between culture stage and ratio of cell abundance between the *Peptococcaceae* OTUs with the rest of the community. The abundance axis has a square-root scale, and points are jittered relative to the culture stage axis. Note that OTU685366684 was correlated with stage 1 and identified as *Pseudomonas aeruginosa* species, a taxonomy not strongly represented in the bioreactor. **c** The succession in time of the community structure in relationship with the culture stage is shown, with the usage of UMAP and Bray-Curtis dissimilarity on relative cell abundance matrix. Note that the most diverse community structures are observed in the bottom left cluster of the panel, which includes points that belong to stages 1 and 2 throughout all time points of succession (day 0 to 34). A smaller, denser cluster in the top exhibit a similar trend. The cluster in right is composed of cultures from stage 2 and 3 (mid and late phase of the succession). Based on PERMANOVA, the community structures were found significantly different based on the three culture stages (pseudo-$R^2$: 0.268 with *p*-value: 0.001).

15 years with benzene as main source of carbon and energy and nitrate as electron acceptor. In addition, the feed contains some vitamins as minor carbon source and ammonium as minor energy source. The metabolic potential and gene expression activity of the MAGs that make part of the microbial community suggest niche partitioning[38] with seven representative dominant members, all of which we will discuss below. We hypothesize that the organisms represented by those MAGs interact with each other via syntrophy, scavenging, predation or even cheating (Fig. 5). Maintenance of and interactions within this relatively complex microbial community may be envisaged in this type of bioreactor where a continuous flow system is in contact with the biofilm. As such, it may host species that grow at much lower rates than the dilution rate[9,39]. Also, the biofilm may have a high potential for spatial heterogeneity, further contributing to a

higher biodiversity[40,41]. In addition, it may confer functional stability since it can shield important functional groups from disturbances and as such may be less prone to invasion by taxa that could interfere with community functioning[39,42,43]. The high biodiversity sustained in the bioreactor over such long timescales ensures metabolic plasticity of the microbial community, which is important for adaptive responses to environmental changes such as the supply of electron donors, electron acceptors, or nutrients. Indeed, the same consortium was shown to readily use iron(III) or sulfate instead of nitrate as the electron acceptor for benzene degradation[9].

We found the highest transcription activity of most of the community members in the biofilm, while only four members of the community were shown to be more active in the liquid phase. Our findings show that only two members of the community are

**Table 1 Taxonomic classification of selected members of the microbial communities in the bioreactor and in the cultures of the succession experiment.**

| OTUs | Phylum | Class | Order | Family | Genus | Species |
|---|---|---|---|---|---|---|
| **OTU624837510** | Firmicutes | Clostridia | Clostridiales | Peptococcaceae | Thermincola | unclassified_Thermincola |
| OTU685366684 | Proteobacteria | Gammaproteobacteria | Pseudomonadales | Pseudomonadaceae | Pseudomonas | Pseudomonas_aeruginosa |
| OTU717462002 | Proteobacteria | Betaproteobacteria | Rhodocyclales | Rhodocyclaceae | unclassified_Rhodocyclaceae | unclassified_Rodocyclaceae |
| OTU91680185 | Firmicutes | Clostridia | Clostridiales | Peptococcaceae | Thermincola | unclassified_Thermincola |

| MAGs | Phylum | Class | Order | Family | Genus | Species |
|---|---|---|---|---|---|---|
| MAG 1 | Chloroflexota | Anaerolineae | Anaerolineales | envOPS12 | UBA7227 | UBA7227 sp002473085 |
| MAG 3 | Planctomycetota | Brocadiae | Brocadiales | Brocadiaceae | Kuenenia | |
| MAG 5 | Chloroflexota | Anaerolineae | Anaerolineales | envOPS12 | OLB14 | |
| MAG 6 | Proteobacteria | Gammaproteobacteria | Burkholderiales | Rhodocyclaceae | UTPRO2 | UTPRO2 sp002840845 |
| **MAG 9** | Firmicutes | Thermincolia | Thermincolales | UBA2595 | | |
| MAG 18 | Bacteroidota | Bacteroidia | Flavobacteriales | koll-22 | | |
| MAG 20 | Bacteroidota | Ignavibacteria | Ignavibacteriales | Ignavibacteriaceae | | |

**Upper table)** Taxonomy assignment of strongly correlated OTUs. The selection was based on random forest variable importance and PERMANOVA analysis. **Bottom table)** The dominant MAGs in the microbial community of the bioreactor. Note that OTU624837510 from the succession experiment and MAG 9 from the bioreactor are marked in bold font as their 16S rRNA sequences are identical.

directly involved in the degradation of benzene. These are members of the *Rhodocyclaceae* (MAG 6) and of the *Peptococcaceae* (MAG 9), the latter of which is the highest active in expressing genes for benzene metabolism. The member of the *Rhodocyclaceae* was found to be more active in the biofilm as compared to the liquid phase. The member of the *Peptococcaceae* on the other hand is one of the four species that is more active in the liquid phase. In line with this observation is the high expression levels of genes encoding the flagellar system, suggesting that *Peptococcaceae* (MAG 9) is actively moving. Additionally, Figs. 1b and 2c show that the two primary consumers of benzene are significantly different from each other in their overall potential functionality. We, therefore, suggest niche partitioning for these two benzene degrading organisms. It was shown previously that a member of the *Peptococcaceae* was an important member of the anaerobic benzene degrading community[9,13,14,28,44]. Our data confirm that hypothesis and we postulate that a member of the *Peptococcaceae* is one of the primary benzene consumers with benzoate/hydroxybenzene, benzoyl-CoA and hydroxypimelyl-CoA as intermediates. To our knowledge, there are no reports on the isolation of a member of the Peptococcaceae with a similar type of metabolism. In our study, we provided an initial characterization of such an uncultivable organism by exploration of MAG 9.

Some members of the Burkholderiales (MAGs 19, 25, 47 and 68) were also classified as benzene consumers. We noticed that they have the potential to make enzymes for degradation of benzoate and hydroxybenzene, two intermediates of benzene degradation, which may diffuse out of the primary consumers and become available to other members[14]. Burkholderiales were described as important species in enrichment cultures of anaerobic benzene-degrading microcosms and were suggested to use the methylation pathway for anaerobic benzene activation[9,45,46]. In our study, we found expression of genes encoding enzymes for benzene degradation that typically combine anaerobic steps with aerobic ones, one of which includes oxygenation to convert benzoyl-CoA to acetyl-CoA and succinyl-CoA. This may seem surprising for an anoxic culture, but the explanation might well be that oxygen is produced locally. Indeed, such an incident production of oxygen has been observed in the anaerobic benzene-degrading microbial community in a previous study (Supplementary Fig. 33)[13]. We hypothesize that such production is achieved by certain species that express a nitric oxide dismutase (NOD), and which may come available to other members of the community. The presence of at least two species that can make NODs in our culture is convincing as their relevant protein sequences show the typical characteristics of a NOD (Supplementary Fig. 34). We envisage the oxygen producers in close

proximity of the oxygen consuming members resulting in very low steady-state levels of oxygen. We further noticed that a gene encoding protocatechuate 4,5-dioxygenase is highly expressed in two different β-proteobacterium (MAGs 19 and 68, successively). The corresponding enzyme catalyzes the oxygen-dependent ring opening of 3,4-dihydroxybenzoate to yield 4-carboxy-2-hydroxymuconate semialdehyde. This observation adds weight to the suggestion that there is local oxygen production in the bioreactor that allows these two species to degrade dihydroxybenzoate aerobically. Moreover, they have also high levels of mRNA encoding benzoyl-CoA oxygenase, which is an oxygen-dependent key enzyme in one of the central branches of benzene degradation. They share this property with yet another three MAGs, 36, 47 and 56. However, despite the precautions[9] we cannot fully exclude the possibility of oxygen contamination.

Other dominant members are exemplified by MAGs 1 and 5 (Chloroflexi), MAGs 18 (Bacteroides) and 20 (Ignavibacterium). Chloroflexi are detected in a wide range of anaerobic habitats where they are highly abundant and seem to play an important role in formation of flocs and biofilms[47–49]. In our culture, all Chloroflexi showed significantly higher expression levels in the biofilm as opposed to the liquid phase. Without exception, they all showed relatively high levels of mRNAs encoding extracellular peptidases, solute binding proteins and specific ABC-type transporters. This observation suggests that they grow by cutting extracellular peptides and proteinaceous polymers into smaller molecules that can be transported into the cell to use them as carbon and energy sources. Other species (MAGs 12 and 49) appear to focus on the synthesis of enzymes for fatty acids uptake and metabolism (Supplementary Section 2.3). We therefore speculate that cell lysis of other members of the community may result in the release of these macromolecules in the bioreactor. Such lysis may occur by bacterial members of the community that express at least one of the types II, III or VI secretion systems, the genes and mRNAs of which we identified in some of the MAGs (Supplementary Fig. 9). Secretion systems induce cell death by the introduction of toxins and other effector molecules in the host cell[50–52]. Perhaps this type of predation is another strategy for some members of the community to occupy their own niche and to survive as secondary consumers. In other studies, members from the Bacteroidetes were regarded as putative biomass scavengers during syntrophic breakdown of benzene and to express genes for a type VI secretion system[29,53]. The latter does not seem to be the case for the dominant Bacteroides in our bioreactor (MAG 18) as the annotation pipeline did not yield a secretion system for this MAG. Instead, it rather behaves as the Chloroflexi in the sense that it expresses extracellular peptidases and peptide

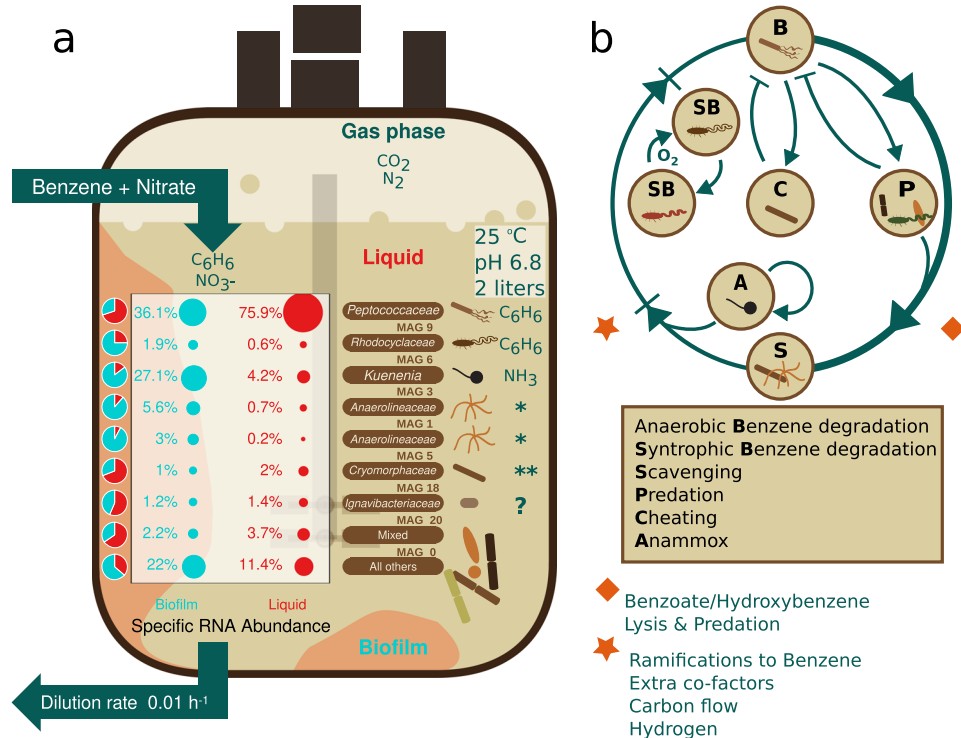

**Fig. 5 Schematic representation of the anaerobic benzene degrading microbial community in a 15-year old bioreactor, highlighting the dominant niches with their hypothetical interactions. a** Characteristics of the 7 dominant MAGs concerning expression levels in biofilm and liquid along with their preferred energy sources are shown. Pie charts show the ratio of relative specific RNA abundance between biofilm (teal) and liquid (red) with the corresponding values. Values and symbols in teal are derived from samples of the biofilm phase, those in red from samples of the liquid phase. The terms biofilm and liquid phases are explained in the Methods section. Note that most expression in both phases is derived from anaerobic benzene degradation by a member of the *Peptococcaceae*, while that of anaerobic ammonia oxidation by *Candidatus* Kuenenia stuttgartiensis is highest in the biofilm. (* - Lysate) *Anaerolineaceae* potentially using extracellular peptides and proteinaceous polymers derived from lysed cells as carbon and energy source. (**) *Cryomorphaceae* potentially using secondary metabolites derived from anaerobic benzene degradation. (?) prevalent carbon and energy source unknown. **b** The outer ring represents the main carbon flow between the primary consumer **B** and the other members of the community. The diamond symbol indicates mechanisms or metabolic left-overs with which **B** feeds the other members of the community and the rhombus symbol indicates what it gets in return. We hypothesize that the community members interact with each other via syntrophy, scavenging, predation or even cheating. Those interactions are represented with the arrows that indicate transition and bar-headed arrows to specify inhibition—in or within the outer ring. The arrow that returns back to **A** represents the autotrophic niche, which is occupied by Anammox for free energy transduction and carbon fixation.

uptake systems. Moreover, it expresses enzymes for the anaerobic degradation of benzoyl-CoA. The reason for the dominance of MAG 20 is unknown.

Another important niche within the community is occupied by a member of the Planctomycetes (MAG 3), which is an autotrophic organism that uses anammox for free energy transduction and carbon fixation[54]. Genes encoding subunits of the key enzyme of anammox, hydrazine synthase, show a high similarity with those from *Candidatus* Kuenenia stuttgartiensis[54]. As such, it is unique in its choice for the energy source. This autotrophic species is known to have a low specific growth rate but it is maintained in the bioreactor by nestling in the biofilm as judged by the allocated expression activity. Then we noticed a few other MAGs with the potential to occupy yet other niches using unique types of metabolism. We found upregulated expression of a cluster of genes encoding enzymes for methylamine metabolism in MAG 16, which belongs to the α-proteobacteria. Methylamine is a C1-compound that is formed during decomposition of proteins and may be taken up specifically by this methylotrophic organisms to use it as nitrogen, carbon and energy source[55]. Another α-proteobacteria (MAG 4) has high levels of mRNA expressed from a cluster of genes encoding enzymes for formate metabolism.

The succession experiment and downstream correlation analyses not only identified the drivers of the community, but also

the assignment of 3 different culture stages with regard to benzene and nitrate consumption rates. Only the cultures that make part of stage 2 or stage 3 reduce nitrate and degrade benzene, in parallel with a lower production rate of nitrite. We postulate that stage 1 is dominated by consumption of the acids and vitamins along with an unbalanced reduction of nitrate into nitrite and further. We then investigated which OTUs are positively correlated with each of the three culture stages, which could cause the transition of the community composition. The OTUs belonging to the family of *Peptococcaceae* showed the strongest correlation, followed by an OTU that belongs to the family of *Rhodocyclaceae*. These two are also the most likely primary consumers of benzene in the bioreactor. A partial confirmation of this identity is that the 16S rRNA sequence of the most abundant OTU was found to be identical to the one of the *Peptococcaceae* in MAG 9 from the original bioreactor.

Overall, our integrative systems ecology approach revealed that many different niches are occupied in the anoxic benzene degrading bioreactor. We hypothesize that niche partitioning in turn results in a community with a relatively high biodiversity (Fig. 5a). Only a few species appear to metabolize benzene or breakdown products thereof. Yet, a wide range of microorganisms does not seem to feed on benzene but most likely on intermediates of benzene degradation, on biomolecules of microbial

necromass, or autotrophically with ammonium as energy source. As a result, most of the community members make up a specialized food web with different trophic levels despite the limited resources.

## Methods

**Metagenomics sequencing.** We selected three samples to use for total DNA high-throughput sequencing. Two of these samples were from brown biofilm (denoted Biofilm 3 and Biofilm 4) collected directly from the bioreactor (Supplementary Section 1.9 for details) and a third sample (denoted BATCH) was produced by pooling equimolar concentration of DNA from batch cultures prepared in triplicate and inoculated separately. The metagenome libraries using Biofilm 3, Biofilm 4 and BATCH samples were prepared using total DNA extraction for subsequent cluster generation and DNA sequencing using the low-throughput Illumina TruSeq DNA Sample preparation Kit (Illumina, San Diego, California, USA) following instructions from the manufacturer. Later, the libraries were analyzed in a Bioanalyzer 2100 (Agilent Technologies, Santa Clara, CA, USA) and diluted to approximately 8 pM with the addition of 5% PhiX and sequenced using two runs in a Illumina HiSeq 2500 platform (Illumina, San Diego, California, USA) according to the instructions of the manufacturer. The BATCH metagenome was sequenced at the user sequencer facility at the VUmc (Amsterdam, The Netherlands). Biofilm 3 and Biofilm 4 metagenomes were sequenced at GATC-Biotech, Konstanz, Germany.

**Metagenomics analysis.** Adapter cutting and quality assessment were performed by Trim Galore v0.4.0[56], a wrapper for Cutadapt[57] and FastQC[58]. Bases with a Phred score below 20 were cut off. The paired option was applied with the standard length cut-off of 20 bases, removing a read pair when one or both of the reads is shorter than 20 bases after quality trimming. We used the most diverse and deeply sequenced sample Biofilm 4 for initial assembly. Reads passing the quality assessment were assembled with IDBA_UD v1.1.1 under standard parameters for short reads with the exception that the maximum kmer size was set at 160[59]. Contigs resulting from the assembly were represented by 93.96% of the total reads and were placed into 111 MAGs with MaxBin v2.1.1 under standard settings and using quality assessed reads from three sequenced samples[60]. MaxBin is dependant on several programs of which the following versions were installed: FragGeneScan v1.20[61], Bowtie2 v2.2.6[62] and HMMER3 v3.1b1[63]. For each MAG, genes were predicted with Prodigal v2.6.2 for single genome parameterization[64], followed by functional annotation of the amino acid sequences of eggNOG-mapper v1.0.3[65]. After a first round of downstream analysis, we refined the MAGs derived from MaxBin using the Anvi'o v6.1 metagenomics workflow[33]. From the refined collection we kept 51 MAGs with completion above 90% and redundancy less than 10%, using the Anvi'o reported scores. Exceptions were made on MAGs 13, 35, 37, 58 and 59, which initially passed the completion/redundancy criteria but later were found highly mixed based on taxonomical classification of their contigs during the manual refinement step. Therefore, they were excluded for further analysis. Differently, MAG 3 (completion: 98.5 %, redundancy: 12.6 %) was included for further analysis, leaving finally 47 high-quality MAGs. Overall, from the reads aligned to the final assembly, 80% is represented in all the reconstructed MAGs, while 48.5% is represented by the final 47 selected MAGs and 30.5% is represented by the 7 dominant MAGs. The final taxonomy was assigned with GTDB-Tk v1.0.2[66], as well with alignment search of the rRNAs genes identified in the MAGs (Supplementary Section 1.10 for details). From the eggNOG annotations, we extracted the KEGG orthology[67] for each MAG and a feature matrix $K$ was constructed of dimensions $k \times m$ where $m$ is the number of MAGs and $k$ is the number of KOs. The entries $K_{qj}$ are 1 if the KO $q$ is present in MAG $j$ and 0 otherwise. Overall 6449 unique KOs was found between the selected MAGs. We used the matrix $K$ as a measure of indication of potential functionality for the microbial community.

**Multi-omics analysis.** We used in total five out of six metatranscriptomes previously obtained from bioreactor samples and described by our collaborators[13,14,68] (Supplementary Section 1.11 and Supplementary Table 6 for details). All five samples were gathered by scraping off a confined area of thick parts of the biofilm or thin ones, originally referred to as brown and white biofilm, respectively. The sixth sample was a partial mixture of both liquid and biofilm phases and not included for further analyses. The two brown biofilm samples (B3 and B4) were estimated to contain around 80% biomass, while the white biofilm samples (E1, E2 and E3) contained around 10%. Brown and white biofilm samples were therefore regarded as biofilm and liquid phase, respectively. The mRNA reads were mapped against the predicted genes of each MAG using bowtie v2.3.4.1, samtools v1.2.1[69], and assigned together with eggNOG annotations. The DNA Abundance of MAGs was calculated by Anvi'o workflow[33]. For RNA Abundance (Eqs. (1) and (2)) we denoted each MAG by $j \in \{1...m\}$, each gene in MAG $j$ by $i \in \{1...n_j\}$ and each sample by $s \in \{1...5\}$ where $1...2$ are samples from the biofilm and $3...5$ are samples from the liquid phase. Then, we defined $r_{jis}$ as mRNA read counts per MAG, gene and sample, and $Tr_s$ as the total mRNA read count of a sample. The matrix $RNAAbundance_{jp}$ resulted from the concatenation of RNA abundance of the column vectors of biofilm and liquid: $B_j$ and $L_j$, respectively.

$$Tr_s = \sum_{j=1}^{m}\sum_{i=1}^{n_j} r_{jis}, \quad B_j = \sum_{i=1}^{n_j}\frac{\sum_{s=1}^{2} r_{jis}}{\sum_{s=1}^{2} Tr_s}\times 10^6, \quad L_j = \sum_{i=1}^{n_j}\frac{\sum_{s=3}^{5} r_{jis}}{\sum_{s=3}^{5} Tr_s}\times 10^6 \quad (1)$$

$$\text{RNA Abundance}_{jp} = \begin{bmatrix} B_j L_j \end{bmatrix} \quad (2)$$

where $p$ indicates the phase, biofilm or liquid ($p \in \{b, l\}$). To obtain a final measure of potential activity, we normalized the RNA abundance by the total genome size per MAG into specific RNA Abundance (Eq. (3)). Therefore, if $G_j$ corresponds to genome size of a MAG $j$ and the overline symbol (like $\overline{G}$) indicates the average taken over all MAGs (similarly $\overline{Tr}$ and $\overline{Mr}$ indicate averages below), then:

$$\overline{G} = \frac{\sum_{j=1}^{m} G_j}{m}, \quad \text{specific RNA Abundance}_{jp} = \frac{\text{RNA Abundance}_{jp}}{G_j}\times \overline{G} \quad (3)$$

To calculate the percentage of active ORFs ratio, we consider an ORF to be potentially active if the average value of the mapped mRNA reads across all samples was equal or bigger than 1. For the differential transcribed analysis of the MAGs we calculated matrix $\Delta$RNA Abundance$_{js}$ (Eq. (4)) for all samples. We donated $r$ and $Tr$ same as above and $Mr$ as the total mRNAs per MAG, then:

$$Mr_j = \sum_{i=1}^{n_j}\sum_{s=1}^{5} r_{jis}, \quad r'_{jis} = \frac{r_{jis}}{Tr_s}\times \overline{Tr}, \quad \Delta\text{RNA Abundance}_{js} = \sum_{i=1}^{n_j}\frac{r'_{jis}}{Mr_j}\times \overline{Mr} \quad (4)$$

Linear modeling, empirical Bayes moderation and multiple testing with Benjamini–Hochberg method were used to validate the differences at mRNA expression[70,71]. For the analysis of DNA, RNA, $\Delta$RNA and specific RNA abundance the log2 scale was used. We used R-packages apcluster, uwot and boruta for downstream clustering, dimensionality reduction and importance feature ranking analysis, respectively[72–77] (Supplementary Section 1.12 for details). Also, R packages KEGGREST were used to get KEGG pathway information, ggplot2, ggrepel for visualization and ShortRead for sequence processing[78–81]. Schematic representations and figures were created and polished, respectively, in Inkscape.

**Targeted analysis on anaerobic benzene metabolism.** To investigate anaerobic benzene degradation we devised the theoretical peripheral and central metabolism based on literature information (Supplementary Section 1.13, Supplementary Figs. 10 and 11 for details). For our analysis we simplified the theoretical scheme by selecting the KEGG reactions and the corresponding KOs. This led to 12 custom-made pathways (Supplementary Section 1.14 for details). We calculated a specific RNA abundance per gene $i$ and MAG $j$ as in Eq. (3) (but without averaging over the genes) and added this value to each entry of the $K$ matrix (Methods "Metagenomics analysis" above), when gene $i$ was assigned to KO $q$. The resulting matrix we denote by $KT$ (Eq. (5)):

$$KT_{qj} = K_{qj} + \langle\text{specific RNA Abundance}_i\rangle_j \quad (5)$$

$KT$ was used for further analysis on global metabolism, structural components, construct the custom-made pathway selections and visualization of graph/ networks[82] (Supplementary Section 1.14 for details). We used classification scheme on the custom-made pathways (Supplementary Section 2.4 for details) to assign MAGs as potential primary consumer (High/Medium/Low/None classes) of benzene. Finally, we performed custom searches on functions of interest with blast+[83]. To do so, we used "makeblastdb" and "blastp" with default parameterization, custom filtering was applied (identity range % > (50–80) & alignment length above half of targets sequence length) to obtain the best hits.

**Succession of microbial communities.** A succession study in batch cultures was carried out to determine important species as drivers for anaerobic benzene degradation. Before performing the succession experiment the benzene-degrading microbial community from the bioreactor was adapted to batch growth as follows. Under anoxic conditions, 0.5 mL of the original bioreactors culture was inoculated in 100 mL serum bottles containing 50 mL of the same phosphate and bicarbonate-buffered medium as used in the bioreactor, and containing 100 μM benzene and excess of nitrate (4.7 mM) as electron acceptor (Supplementary Section 1.1 for medium composition). After benzene was consumed (Supplementary Section 1.2 for benzene determination), the medium was spiked again with 100 μM benzene. Once three times benzene were completely depleted, the culture was transferred to fresh medium under anoxic conditions (Supplementary Section 1.1). The succession experiments were performed after three such transfers (over a course of 6 months in total) in 50 mL batch culture. The medium used in the succession experiment was the same medium as in ref. [9]. It consists of a mixture of basal salt medium, phosphate solution, carbonate solution, trace elements solution, vitamin solution, nitrate as electron acceptor, and 100 μM benzene. The serum bottles had a 90:10 N$_2$: CO$_2$ (v/v) atmosphere, and were sealed with Viton stoppers and capped with aluminum crimps. The cultures were incubated at 25°C in the dark (Supplementary Section 1.1 for details). For the succession experiment, a total of 96 anoxic serum bottles with 100 μM benzene were inoculated with $2 \times 10^4$ cells per mL at the same time. We determined cell numbers by separating cell aggregates by gentle sonication (3 × 30 s at 15 microns of amplitude with 30 s intervals each), followed by cell staining using SYBR-Green II, and by counting cells in a Accuri C6 Flow Cytometer System (Accuri Cytometers, Ltd., Cambridge, UK)

(Supplementary Section 1.3 for details). The following cultivation and counting controls were performed: (i) no cells added, (ii) no benzene added and (iii) no vitamins added. We prepared five individual bottles for each control (15 in total). The cultures (including the 15 controls) were set up in the 30 mL serum bottles with 20 mL medium. All other conditions kept the same as the adaptation step (mentioned above).

A pilot experiment was performed to determine general trends of benzene consumption over time by taking quadruplicate samples under the same conditions of the final succession experiment. Based on the pilot experiment, we inoculated 96 bottles from which we sacrificed 12 bottles each at time points 2h, 7 days, 14 days, 18 days, 22 days, 26 days, 30 days and 34 days after inoculation. The controls were sampled 34 days after the beginning of the experiment. Total DNA was extracted from each of the 96 samples and the 5 controls using a modified CTAB/phenol-chloroform described previously[84] (Supplementary Section 1.4 for details). DNA from 9 bottles at time point 2h were too low, leaving the DNA from 87 experimental samples for further analyses. The DNA was used for 16S rRNA gene amplicon sequencing including negative controls (i.e. no addition of template DNA to the PCR). We analyzed all samples when sacrificed for benzene by gas chromatography and cell number by flow cytometry (Supplementary Sections 1.2 and 1.3 and Supplementary Data 2 for details). We also filtered two times 2 mL of each cell suspension through 0.22 µm nitrocellulose filter membranes (Merck, Darmstadt, Germany). One of the filtrates was used to determine nitrate and nitrite concentrations by capilary electrophoresis (Supplementary Section 1.5 for details) and targeted metabolomics with the corresponding controls. Targeted metabolomics was used to measure the concentration of the different vitamins used in the medium, including biotin and vitamin B12, and potential intermediates of anaerobic degradation benzene degradation by LC-MS/MS (Supplementary Sections 1.6 and 1.7 for details).

**16S rRNA gene high-throughput sequencing**. The 16S rRNA V3-V4 region of the samples from the succession experiment was sequenced using the primers S-D-Bact-0341-b-S-17 and S-D-Bact-0785-a-A-21[85]. To minimize PCR bias, we performed PCR reactions in triplicate for each sample. Due to low cell biomass each 25 µL reaction contained 0.05 µg of DNA (Supplementary Section 1.8 for details). Amplicons from all samples were pooled in equimolar concentrations into one composite sample and were paired-end sequenced at the Vrije Universiteit Amsterdam Medical Center (Amsterdam, The Netherlands) on a MiSeq Desktop Sequencer with a 600-cycle MiSeq Reagent Kit v3 (Illumina, San Diego, California, USA) following instructions of the manufacturer. High-throughput sequencing raw data were demultiplexed and processed using a modified version of the Brazilian Microbiome Project 16S rRNA profiling analysis pipeline[86] (Supplementary Section 1.8 for details). To estimate the total number of species present at the start of the succession experiment we pooled all the sacrificed samples together and identified 192 OTUs that shared more than 97.0% sequence similarity. The OTU abundance was normalized to the total equivalent of cell numbers using the estimated 16S copy number per cell for each OTU. To do so, the ribosomal RNA operon copy number database, hereafter rrnDB[87] was used. These estimates were run using RDP Classifier version 2.10.1 and RDP training set No. 10 incorporating 16S copy number data from the downloadable pan-taxa tables of rrnDB version 4.2.3[88]. The taxonomic assignment for each OTU (Supplementary Section 1.8 for details) was used to link the correspondent rrnDB entry. Further, we normalize the data to the equivalent cell number per mL (OTU-specific cell densities) using the flow cytometry cell counts[89]. Selection of important OTUs, which correlated with the three stages of the culture development, was performed with random forest implementation of R-package party[90]. We used default parameters for conditional variable importance, which uses permutation on mean decrease in accuracy. For that, 1000 trees and 10 numbers of random sampling were set. Also, we used R-package vegan implementation of Permutational Multivariate Analysis of Variance (PERMANOVA) on 999 permutations[91–93].

**Statistics and reproducibility**. All the details considering the statistical tests that we applied, the sample size, and the type of replicates are presented in each relevant section of the methods as well as in the supplementary information.

**Reporting summary**. Further information on research design is available in the Nature Research Reporting Summary linked to this article.

## Data availability
Processed data/tables can be found in: https://github.com/SystemsBioinformatics/anaerobic-benzene-degrading-culture. Metagenomics shotgun & 16S rRNA gene amplicon sequencing are available in the European Nucleotide Archive under primary accession PRJEB39357. Metagenome-assembled genomes can be found in an Anvi'o result at zenodo open-access repository under https://doi.org/10.5281/zenodo.3939224[94].

## Code availability
Code can be found in: https://github.com/SystemsBioinformatics/anaerobic-benzene-degrading-culture[95].

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

## Acknowledgements

We thank Laura Luzia, Frank Bruggeman, Bas Teusink and Bastian Hornung for their significant contribution with discussions and proofreading. Evelina Tutucci, Paul Iturbe Espinoza, Philipp Savakis and Huub J. M. Op den Camp for helpful discussions. We are grateful to the late Wilfred F.M. Röling for his scientific input on experimental design. Unfortunately, he passed away before this work was published. This study was supported by a grant of BE-Basic-FES funds from the Dutch Ministry of Economic Affairs. The research of CM is supported by a Grand Solution grant from Innovation Fund Denmark (grant no. 6150-00033B), The FoodTranscriptomics project.

## Author contributions

C.M. and D.M. conceived the methodology, wrote the code and performed the analysis. Succession experiment and flow cytometry was performed by L.F., M.B. and U.N.d.R. Metabolomics was performed by L.F., R.H. and J.P. Nucleic acid analysis was performed by U.N.d.R. and E.K. B.B. contributed to the 16S rRNA sequence data processing and analysis. M.v.d.W. and J.G. conceived and conducted the continuous culture experiment. S.A. and H.S. performed metatranscriptomic analysis. L.F., R.v.S. and U.N.d.R. theore- tical reconstruction of metabolic pathways. Initial computational metabolic pathway reconstruction was done by W.G. & B.O. W.F.M.R. contributed on the experimental design. C.M. & R.v.S. wrote the paper. All authors reviewed the manuscript.

## Competing interests

The authors declare no competing interests.
