## [Peer Review File · Communications Biology]

Reviewers' comments:

Reviewer #1 (Remarks to the Author):

Melvokonian et al studied a long-term (15 years) benzene-degrading flow-through enrichment culture, and for its microbial compositions and functioning using metagenomics including a 111 MAGs and transcriptomics. Metagenomic coverage and transcription of the MAGs were correlating. The study compares the microbial community compositions in a biofilm with the planktonic fraction. The authors found that next to the benzene degraders a complex network of microorganisms was established that thrives on metabolites, necromass etc. Batch cultures produced from the long-term culture were also studied.

Fig. 1 shows that the difference between biofilm and liquid communities, and defines 3 functional groups that are further studied. Fig 2 uses heat maps to compares function of MAGs in Benzene degradation for two organisms and the potential of organisms from dominating and non-dominating groups in alkane degradation. Fig. 3 finally looks at the metabolism in a batch culture that was set up from the long-term culture, and defines three stages. Fig. 4. displays the relative abundance of different OUTs (now 16S tag seq) in the 3 culture stages defined in Fig.3.

As summary the study shows the dominance of two alkane degraders (Peptococcaceae and Rhodocyclaceae with one anammox bacterium (Kuenenia) and additional, likely heterotrophic organisms that thrive in the bioreactor. The way to achieve that results seems to be overly complicated (including 37 supplementary figures). Though I do not see a specific answer to to real environmental question.

Some more comments:

In general more precise language would be important ,including the correct use of biochemical and microbiological terminology

I am sure more is known on the metabolism of Peptococcaceae

Line 149 ff + discussion: What do we learn from the development of community compositions within that short time frame?

Line 190; The metagenomic analysis is so deep that nitric oxide dismutases should be identified (or search the whole metagenome for it).. It is not likely that oxygen produced by additional community members is sufficient to allow aerobic pathways.) I rather think, if oxygen is used by organisms, then it is a contamination.

Reviewer #2 (Remarks to the Author):

In this paper Melkonian and colleagues investigated a microbial community which evolved in a 15-years old bioreactor with benzene as the main carbon and free energy source and nitrate as the electron acceptor. The study focused on the metabolic pathways and the metatranscriptome in order to identify the community members directly involved in benzene degradation and on the putative role of the other species. By using genome-centric metagenomics combined with metatranscriptomics and analysis of the chemical compounds the Authors obtained a general representation of the microbiome, of the nice partitioning, and they were able to determine how species richness is maintained and how the complexity of a natural community is stabilized. The paper is clear and well-written, results are of interest and very well presented using clear pictures and graphs and a very rich section of supplementary informations.

I think that Authors have to address the comments provided below in order for the manuscript to be eligible for publication. For this reason I provided below some general suggestions for the manuscript improvement together with some more specific comments.

Major comments.

Comment 1-I was not able to find in the manuscript information describing the submission of the sequences to public databases (Sequence Read Archive or similar). Additionally, I can suggest to submit the MAG sequences to open access resources such as figshare or similar in order for other researcher to be able to inspect the results and re-use the data presented in the manuscript.

Comment 2-Abstract is quite short and schematic, I am not sure the main findings are really communicated to the audience. For example a sentence can be added to describe the approach used (genome-centric metagenomics plus metatranscriptomics) or to provide some information regarding the species (MAGs) identified and their functional role.

Comment. 3. Did Authors try to use similarity search on 16S rRNA genes (or similar approaches) to link the OTUs identified using amplicon sequencing with the MAGs sequences? This can help to track the OTUs to their function and to link the two analyses performed with amplicon sequencing and metagenomics.

Comment 4. I was not able to find in the manuscript a list of genes potentially involved in biofilm formation. Did authors checked the microbes potentially involved in this process? This can be useful to have a better understanding regarding the role of the species in the microbiome functioning.

Comment 5. Did authors verify the fraction of the microbiome represented by the selected OTUs/MAGs? This can be easily obtained by mapping the reads back to the genomes and can be useful to understand if the selected MAGs are describing most of the microbiome or, on contrary, if there is a substantial part of the microbiome that remained unidentified (or represented in the MAGs discarded due to low quality).

Comment 6. The procedure used to partition the MAGs considering their KEGG functions (lines 274-277 and supplementary information 1.12) is very interesting. These approaches are strongly needed by the scientific community and can help to improve the microbiome investigation. Authors can consider to provide a detailed pipeline of the process and made it available through github (or similar platforms) in order to allow other researcher to re-perform the procedure. This can attract more readers and increase the general interest of the paper.

Line 27-Authors can consider to add citation to recent papers published in this topic such as for example "Whu X et al. Microbiome volume 8, Article number: 22 (2020)".

Line 36. If the 16S sequencing was applied, then probably it is more correct to refer to the "number of OTUs".

Line 53. Authors can consider to add citation to recent papers regarding transcriptomics such as "Yangyang et al., Biotechnology for Biofuels volume 11, Article number: 117 (2018)".

Line 69. Authors can provide a more clear explanation regarding the correlation between transcriptional activity and MAGs abundance. Are there significant differences between the two measures? These aspects can be adressed better. Additionally, is it possible that the activity

determined using RNA-seq was influenced by the presence of rRNA genes recovered only for some of the MAGs? In fact most of the RNA-seq reads derive from ribosomal RNA and they map only on MAGs where the rRNA gene(s) have been identified.

Minor comments.

Line 106. What is the similarity level? Is it reported somewhere in the paper? Please provide details regarding figure/tables.

Line 161. There is a typo, please check and correct.

Line 266. Software name is "Prodigal", not "prodigal".

Reviewer #3 (Remarks to the Author):

This work employed metagenomics approach to characterize the potential activities of the microbiome in an anaerobic benzene degrading culture and identified metagenome-assembled genomes (MAGs) that likely involved in those activities. However, I cannot find quantitative evidence that support the authors' arguments that different species (or MAGs) exhibit niche partitioning or interactions. Actually, their findings are basically descriptive, with many results focusing on specific and detailed information regarding functional/taxonomic annotations of MAGs, but the author did not identify niche or interactions of MAGs. Thus, the whole story about "niche partitioning" and "a food web with different trophic levels" is just a hypothesis but not their conclusion. I think the approaches and findings are interesting. However, the authors need to carefully reframe the story to avoid over-emphasize their implications.

I must admit that I am not familiar with metagenomic analyses; and I assumed their molecular methods and bioinformatics are correct. Thus, evaluation by other experts on this part is needed.

In the following, I made some editorial comments that aim to improve the readability of the manuscript.

Title:

The title needs to be more specific.

Abstract

The abstract needs to be more specific. Please explain the evidence that niche partitioning was uncovered from their data and that niche partitioning maintained high diversity and complexity. Specifically, what analyses were done and what results support the claims. As I explained above, I could not find any evidence or statistical support to the statements in Abstract from their Results.

L20: "These mechanisms may well be conserved across ecosystems."

I am not sure what the authors imply here?

Introduction

L53-55: "An additional experiment was designed to independently identify the main organisms that drive anaerobic benzene degradation. To this end, we inoculated a series of batch cultures of the 15 years-old microbial community at low cell densities."

It's unclear why and how the authors test their hypothesis through a series of batch cultures. For example, how do species interact (such as competition for resources, collaboration by exchange of

products, inhibition in a chemical warfare and spatial organization) resulting in the compound patterns like Figure 3. There was no clear explanation in Methods how to quantify interactions and niche partitioning, either.

Results

L70-73: "The majority of the MAGs was found to have a significantly"

This sentence regarding RNA abundance is uninformative. Moreover, I suspect that this result could be biased because the metagenomes (DNA as the template) were derived only from the biofilm, whereas the transcriptomes (RNA) were taken from both the biofilm and the liquid (under-represented in the template).

Before L128 vs. After L128:

The transition and linkage between "results based on metagenome data in the original bioreactor" vs. "results based on the 16S rDNA gene in newly generated batch grown cultures" are required for addressing the research objectives.

Discussion

L166-169: "We hypothesize that they interact with each other via syntrophy, scavenging, predation or even cheating (Fig. 5)....."

Here, in the discussion, the authors finally mention their "hypothesis", with Fig. 5 to summarizing their findings in metagenome-assembled genomes (MAGs). Again, this is their hypothesis but not their findings.

L227-228: "The succession experiment and downstream correlation"

The authors provided succession data. However, instead providing new insights regarding species interactions, these data were mainly used to postulate the compound consumption in different stages and support the results of MAGs (primary consumers of benzene members were OTUs belonging to the family of Peptococcaceae (MAG 9) and Rhodocyclaceae (MAG 6) in the bioreactor). Thus again, the authors did not analyze interactions and niche partitioning.

L236-246: "Overall, our integrative systems ecology approach....."

The same as the context in the Abstract, the whole story about "niche partitioning" and "a food web with different trophic levels" seems only according to the genome-centric hypothetical scheme (Fig. 5). This is hypothesis but not their finding.

We would like to thank all reviewers for thoroughly reading and assessing our work. We noticed that it improved the quality of our paper substantially. We will address the points raised by the reviewers item by item below. Text with blue color indicates the reviewers points and text in black color our responses. All changes (such as addition, deletion or replacements) below as well in the updated manuscript are indicated with the green color.

Referee expertise:

Referee #1: microbial hydrocarbon metabolism, biochemistry and molecular ecology

Referee #2: broad interests in genomics and metagenomics.

Referee #3: oceanographer, application of ecological theory using statistical theoretical tools.

Reviewers' comments:

Reviewer #1 (Remarks to the Author):

Melvokonian et al studied a long-term (15 years) benzene-degrading flow-through enrichment culture, and for its microbial compositions and functioning using metagenomics including a 111 MAGs and transcriptomics. Metagenomic coverage and transcription of the MAGs were correlating. The study compares the microbial community compositions in a biofilm with the planktonic fraction. The authors found that next to the benzene degraders a complex network of microorganisms was established that thrives on metabolites, necromass etc. Batch cultures produced from the long-term culture were also studied.

Fig. 1 shows that the difference between biofilm and liquid communities, and defines 3 functional groups that are further studied. Fig 2 uses heat maps to compares function of MAGs in Benzene degradation for two organisms and the potential of organisms from dominating and non-dominating groups in alkane degradation. Fig. 3 finally looks at the metabolism in a batch culture that was set up from the long-term culture, and defines three stages. Fig. 4. displays the relative abundance of different OUTs (now 16S tag seq) in the 3 culture stages defined in Fig.3.

As summary the study shows the dominance of two alkane degraders (Peptococcaceae and Rhodocyclaceae with one anammox bacterium (Kuenenia) and additional, likely heterotrophic organisms that thrive in the bioreactor. The way to achieve that results

seems to be overly complicated (including 37 supplementary figures). Though I do not see a specific answer to a real environmental question.

We have added an extra sentence to the introduction to clarify the environmental question at L 51-52.

“The aim of this work was to get a more fundamental understanding of the diversity, structure, metabolic potential and dynamics of the anaerobic benzene-degrading microbial community. More specifically, we aimed at getting insight in the driving forces behind the persistence of large biodiversity in natural environments.”

The answer to that question is partly explained by our metagenomics and transcriptomics data along with our hypothesis about niche partitioning in a food web with different trophic levels

Some more comments:

In general more precise language would be important, including the correct use of biochemical and microbiological terminology

It is not completely clear what is meant by the reviewer. We and our colleagues went through the manuscript to identify incorrect use of biochemical and microbiological terminology, but we couldn't find these, although we have extended expertise in these disciplines in the team. We did, however, adopt the term 'necromass' mentioned by the reviewer, as this seems to be the proper term for the contents of dead cells.

I am sure more is known on the metabolism of Peptococcaceae

Most members of the family of Peptococcaceae and, more importantly, of the genus Thermincola are unculturable and therefore not prone to physiological studies. This includes our species as well as our colleagues from Utrecht who have the bioreactor for 15 years were unable to get it in isolation. Therefore, its metabolism can only be deduced from -omics studies. That is what we tried to do in our current study (L 617-621, Code & Data availability). There is a member, though, which is more distantly related, but its metabolism is completely different from our species in that it consumes CO rather than benzene and derivatives in our case. Also their environmental surroundings and optimal temperatures are different. Information we gathered from the paper below:

Clostridia; Clostridiales; Peptococcaceae; Thermincola; Thermincola carboxydiphila gen. nov., sp. nov., a novel anaerobic, carboxydophilic, hydrogenogenic bacterium from a hot spring of the Lake Baikal area .

<https://www.microbiologyresearch.org/content/journal/ijsem/10.1099/ijms.0.63299-0#tab2>

To address this, we added the following phrase at discussion L 197-198: “To our knowledge, there are no reports on the isolation of a member of the Peptococcaceae with a similar type of metabolism. In our study, we provided an initial characterization of such an uncultivable organism by exploration of MAG 9.”

Line 149 ff + discussion: What do we learn from the development of community compositions within that short time frame?

We learned that the development of the community composition in each culture strongly correlated with benzene degradation (Figure 4 c) and by the significant increase of Peptococcaceae MAG 9 (Figure 4 a & b). This finding supports the prediction that MAG 9 is likely the primary consumer of benzene and, hence, the primary producer of biomass for the community (Figure 2, 4 & 5).

We emphasized this lesson specifically in the result text at lines L 165-166: “We subsequently identified a transition in community composition as well in the OTUs of which the cell densities correlated with the three stages of culture development”

And in discussion L 251-252: “We then investigated which OTUs are positively correlated with each of the three culture stages, which could cause the transition of the community composition.”

Line 190; The metagenomic analysis is so deep that nitric oxide dismutases should be identified (or search the whole metagenome for it).. It is not likely that oxygen produced by additional community members is sufficient to allow aerobic pathways.)

I rather think, if oxygen is used by organisms, then it is a contamination.

First, we need to emphasize that we did find nitric oxide dismutases in the whole metagenome and could even identify them in 2 of our MAGs. That is stated in the text (L 110-111 & L 209-210), along with our arguments that NODs have a characteristic pattern of amino acids around the heme binding site that is different from nitric oxide reductases, which we also illustrated in the supplement (Suppl Figure 37). Our experienced colleague Jan Gerritse who maintains the bioreactor for more than 15 years took all the following precautions to avoid oxygen contamination:

- 1) N₂/CO₂ gas was stripped of O₂ using hot copper shreds.
- 2) The reactor was made of glass with a steel lid.
- 3) Media and gas were supplied using Viton tubing, with Norprene pumping tubes.
- 4) Reservoir and reactor headspace were continuously flushed with N₂/CO₂
- 5) The reactor was wrapped in aluminum foil to avoid light and algal growth.

(That information is provided in our supplementary material as well in van der Zaan BM, et al. Environmental Microbiology. 2012
<https://doi.org/10.1111/j.1462-2920.2012.02697.x>.)

Moreover, they also unambiguously demonstrated that oxygen is produced under certain conditions when the culture is spiked with nitrite. This has been published by Atashgahi et al and we refer to this study in the manuscript (L 206-207).

However, we agree with the reviewer that oxygen contamination can never be fully excluded. We include the following phrase in L 217-218. “However, despite the precautions (van der Zaan BM, et al. Environmental Microbiology. 2012) we cannot fully exclude the possibility of oxygen contamination.”

Then about the likelihood whether the oxygen produced by these oxygen producing bacteria is sufficient for growth of others. When the cells are close enough to one another (which may be expected in a biofilm) then the local oxygen concentration might be high enough to sustain aerobic growth, though the rates of growth would then be depending on the flux of oxygen production. We should not forget that the oxygen flux might be relatively low as carbon and free energy metabolism with oxygen as oxidant is much more efficient than anaerobic growth, hence that increased efficiency may compensate for a lower specific oxygen uptake rate. But quantification of aerobic metabolism is not the scope of our study.

We replaced a text in the following paragraph at supplementary text L 508-515 . “We hypothesize that the oxygen produced by these oxygen producing bacteria is sufficient for the growth of others. When the cells are close enough to one another (which may be expected in a biofilm) then the local oxygen concentration might be high enough to sustain aerobic growth (Davey et. al. (2000), van Tatenhove-Pel et. al. (2020)), though the rates of growth would then be depending on the flux of oxygen production. Further, the oxygen flux might be relatively low as carbon and free energy metabolism with oxygen as oxidant is much more efficient than anaerobic growth, hence that increased efficiency may compensate for a lower specific oxygen uptake rate. “

We cited the following publication Davey et. al. (2000):
[10.1128/membr.64.4.847-867.2000](https://doi.org/10.1128/membr.64.4.847-867.2000), van Tatenhove-Pel et. al. (2020):
[10.1038/s41396-020-00806-9](https://doi.org/10.1038/s41396-020-00806-9)

Reviewer #2 (Remarks to the Author):

In this paper Melkonian and colleagues investigated a microbial community which evolved in a 15-years old bioreactor with benzene as the main carbon and free energy source and nitrate as the electron acceptor. The study focused on the metabolic pathways and the metatranscriptome in order to identify the community members directly involved in benzene degradation and on the putative role of the other species. By using genome-centric metagenomics combined with metatranscriptomics and analysis of the chemical compounds the Authors obtained a general representation of the microbiome, of the niche partitioning, and they were able to determine how species richness is maintained and how the complexity of a natural community is stabilized. The paper is clear and well-written, results are of interest and very well presented using clear pictures and graphs and a very rich section of supplementary informations. I think that Authors have to address the comments provided below in order for the manuscript to be eligible for publication. For this reason I provided below some general suggestions for the manuscript improvement together with some more specific comments.

Major comments.

Comment 1-I was not able to find in the manuscript information describing the submission of the sequences to public databases (Sequence Read Archive or similar). Additionally, I can suggest to submit the MAG sequences to open access resources such as figshare or similar in order for other researcher to be able to inspect the results and re-use the data presented in the manuscript.

In the section Code & Data availability L 617-621 we provide all the relevant information: "Code and processed data/tables can be found in:

<https://github.com/SystemsBioinformatics/anaerobic-benzene-degrading-culture>.

Metagenomics shotgun & 16S rRNA gene amplicon sequencing are available in the European Nucleotide Archive under primary accession PRJEB39357.

Metagenome-assembled genomes can be found in an Anvi'o result at zenodo open-access repository under <http://doi.org/10.5281/zenodo.3939224>"

Comment 2-Abstract is quite short and schematic, I am not sure the main findings are really communicated to the audience. For example a sentence can be added to describe the approach used (genome-centric metagenomics plus metatranscriptomics) or to provide some information regarding the species (MAGs) identified and their functional role.

We modify the abstract with the reviewers' comments in mind. Due to the limitation of 150 words we could not include information regarding the species (MAGs) identified and their functional role. The current version is 162 words.

“A key question in microbial ecology is what the driving forces behind the persistence of large biodiversity in natural environments are. We studied a microbial community with more than 100 different types of species which evolved in a 15-years old bioreactor with benzene as the main carbon and free energy source and nitrate as the electron acceptor. We demonstrate by using genome-centric metagenomics plus metatranscriptomics that most of the community members most likely feed on metabolic left-overs or on necromass, while only a few of them are candidates to degrade benzene. We verify with an additional succession experiment using metabolomics and metabarcoding that these few community members are the actual drivers of benzene degradation. As such, we hypothesize that high species richness is maintained and the complexity of a natural community is stabilized in a controlled environment by the interdependencies between the few benzene degraders and the rest of the community members, ultimately resulting in a food web with different trophic levels.”

Comment. 3. Did Authors try to use similarity search on 16S rRNA genes (or similar approaches) to link the OTUs identified using amplicon sequencing with the MAGs sequences? This can help to track the OTUs to their function and to link the two analyses performed with amplicon sequencing and metagenomics.

Indeed, we tried to do such a mapping with different approaches. Some of the results are presented in supplementary Table 2 (title “Between experiments”). Additionally, we reported these on results L 169-170 and L 172-174

Also, Anvi'o results provide predictions of 16S rRNA genes per MAG, which are provided in the zenodo repository (<http://doi.org/10.5281/zenodo.3939224>). Relevant information is provided on github (<https://github.com/SystemsBioinformatics/anaerobic-benzene-degrading-culture>): files table_mapped_silva_bins_strict.csv and table_mapped_OTU_bins_strict.csv in folder bioreactor>Data

Comment 4. I was not able to find in the manuscript a list of genes potentially involved in biofilm formation. Did authors checked the microbes potentially involved in this process? This can be useful to have a better understanding regarding the role of the species in the microbiome functioning.

This is a good point, but we did not do that as we did not want to hypothesize too much. Emma Tabe Eko Niba et al (2007) (<https://doi.org/10.1093/dnares/dsm024>) describe studies on *E. coli* showing that (a combination of) more than 110 genes are involved in biofilm formation, most of which have also roles in movement (flagella), fimbriae, LPS

formation, and regulation. What the key genes are is also not clear as Rossi et al (2017) review (<https://doi.org/10.1080/1040841X.2017.1303660>) that, as an example, type 1 fimbriae are important for biofilm formation in the laboratory *E. coli* K-12 strain, but not in *E. coli* EHEC O157:H7. Our view is that such a list would be too speculative to shed light on the role of the species in biofilm formation. We did, however, investigate flagella and pilus systems as illustrated by supplementary Figures 9 and 10. This analysis sheds some light on movement and adhesion within the community, which may be helpful for more targeted research in biofilm formation.

Comment 5. Did authors verify the fraction of the microbiome represented by the selected OTUs/MAGs? This can be easily obtained by mapping the reads back to the genomes and can be useful to understand if the selected MAGs are describing most of the microbiome or, on contrary, if there is a substantial part of the microbiome that remained unidentified (or represented in the MAGs discarded due to low quality).

We verified that 93.96% of the total reads from the two metagenomic samples were aligned at the original assembly. From those reads, 80% is represented in all the reconstructed MAGs, while 48.5% is represented by the final 47 selected MAGs. The 7 dominant MAGs are represented by around 30.5% of those reads.

This calculation also takes into account MAG 000 which contains all contigs which were not binned or were undetermined after the refinement step. These represent around 20 % of the mapped reads.

We included this information in the Methods; Metagenomics analysis section L 284 & L 294-296.

Also the file bins_percent_recruitment.txt is uploaded in the github repository (<https://github.com/SystemsBioinformatics/anaerobic-benzene-degrading-culture>) under bioreactor>Data folder

Comment 6. The procedure used to partition the MAGs considering their KEGG functions (lines 274-277 and supplementary information 1.12) is very interesting. These approaches are strongly needed by the scientific community and can help to improve the microbiome investigation. Authors can consider to provide a detailed pipeline of the process and made it available through github (or similar platforms) in order to allow other researcher to re-perform the procedure. This can attract more readers and increase the general interest of the paper.

We agree with the reviewer. That is why we provided a github repository to accompany this study (<https://github.com/SystemsBioinformatics/anaerobic-benzene-degrading-culture>). An additional github repository is provided (<https://github.com/SystemsBioinformatics/funciminer>) in an older publication to reconstruct the initial matrices. The corresponding publication is cited at L 54. Although the code is not optimized to be used straightforward as a pipeline, anyone with basic coding experience can reproduce our analysis and reuse it to investigate other microbial communities.

Line 27-Authors can consider to add citation to recent papers published in this topic such as for example “Whu X et al. *Microbiome* volume 8, Article number: 22 (2020)”.

We found the suggested article relevant as the reviewer suggested and cite it.

Line 36. If the 16S sequencing was applied, then probably it is more correct to refer to the “number of OTUs”.

The reviewer is right. We changed ‘more than 100 different types of species’ into “more than 100 different types of operational taxonomic unit (OTU)”. L 37-38.

Line 53. Authors can consider to add citation to recent papers regarding transcriptomics such as “Yangyang et al., *Biotechnology for Biofuels* volume 11, Article number: 117 (2018)

We found the suggested article relevant as the reviewer suggested and cite it.

Line 69. Authors can provide a more clear explanation regarding the correlation between transcriptional activity and MAGs abundance. Are there significant differences between the two measures? These aspects can be adressed better.

There are indeed significant differences between the two measures, but the interpretation is complicated. This is because the extraction procedures were not quantitative. Overall, there are positive correlations, yet the exact biological ratios cannot be determined precisely as a consequence of the extraction procedure that we used. Therefore, we have put the plots regarding these findings in Fig. 1 of the supplement and description of that in the text below that Figure. The Figure is also updated to match with *Communications Biology's* suggestions as well with the rest of the work.

The updated Figure description:

“Relation of MAGs RNA abundance with DNA abundance (biofilm samples). a) for RNA derived from the biofilm. b) for RNA derived from the liquid phase. The color of the points indicates the quality of the MAGs. The positive correlation varies from moderate to weak, respectively (biofilm: $r^2 = 0.556$, liquid: $r^2 = 0.286$). The most transcribed MAGs, 3 and 9, were not the most abundant ones as judged by their DNA. Instead, MAG 1 and MAG 2 were the most abundant in DNA content. Specifically MAG 1 was represented in the biofilms community with 25% relative DNA abundance but only with 5.6% relative RNA abundance, suggesting that they are relatively inactive or dead. Alternatively, biases during the DNA extraction cannot be excluded as it is illustrated for example in various studies that the abundance of gram-positive bacteria may be underestimated due to the difficulty of breaking their thicker cell wall \cite{Roopnarain2017}.”

Additionally, is it possible that the activity determined using RNA-seq was influenced by the presence of rRNA genes recovered only for some of the MAGs? In fact most of the RNA-seq reads derive from ribosomal RNA and they map only on MAGs where the rRNA gene(s) have been identified

The reviewer is right to say that there are biases of RNA-seq which may have an influence on the putative activity. This effect is expected to be stronger on low abundant MAGs. After the RNA-seq quality control we also filtered the ribosomal RNA with the usage of SortMeRNA tool, to focus only on mRNAs. To address the reviewers point we investigated the relationship between MAGs transcribed KOs and the total KOs count in supplementary Figure 5 and found that only from the 7 most abundant (later named as dominant) MAGs we assigned mRNAs to all their corresponding KOs.

We also added a supplementary table 2 with the RNA read processing. The table shows the percentage of the final mRNA reads mapped to the MAGs, after quality filtering and rRNA exclusion. The enumeration of the other supplementary tables was updated.

Minor comments.

Line 106. What is the similarity level? Is it reported somewhere in the paper? Please provide details regarding figure/tables.

The average Nucleotide Identity (ANI) between MAG 3 and *Candidatus* Kuenenia stuttgartiensis is 94.7 % and the relative evolutionary divergence (RED) to the genus Kuenenia is 0.98. We included the first value in the revised main text, L 113-114.

A general table of the taxonomy assignment is provided in supplementary table 2. Furthermore, a more detailed information is provided in the github repository (<https://github.com/SystemsBioinformatics/anaerobic-benzene-degrading-culture>) folder bioreactor>Data file gtdbtk.bac120.summary.tsv.

Line 161. There is a typo, please check and correct.

The typo is corrected and updated to “ genome-centric metagenomics combined with metatranscriptomics” L 176.

Line 266. Software name is "Prodigal", not “prodigal”.

The name is corrected (L 287).

Reviewer #3 (Remarks to the Author):

This work employed metagenomics approach to characterize the potential activities of the microbiome in an anaerobic benzene degrading culture and identified metagenome-assembled genomes (MAGs) that likely involved in those activities. However, I cannot find quantitative evidence that support the authors' arguments that different species (or MAGs) exhibit niche partitioning or interactions. Actually, their findings are basically descriptive, with many results focusing on specific and detailed information regarding functional/taxonomic annotations of MAGs, but the author did not identify niche or interactions of MAGs. Thus, the whole story about “niche partitioning” and “a food web with different trophic levels” is just a hypothesis but not their conclusion.

I think the approaches and findings are interesting. However, the authors need to carefully reframe the story to avoid over-emphasize their implications. I must admit that I am not familiar with metagenomic analyses; and I assumed their molecular methods and bioinformatics are correct. Thus, evaluation by other experts on this part is needed.

In the following, I made some editorial comments that aim to improve the readability of the manuscript.

We agree with the general criticism of the reviewer to avoid over-emphasizing the implications of our study. Our conclusion is that a rich and diverse bacterial community survives and maintains over the years though with a narrow range of carbon and free

energy sources. We hypothesize that this observation is explained both by community interactions and niche partitioning of the community which results in a food web with different trophic levels. We have changed the text and clearly state that the latter terms refer to our hypothesis.

We updated the introduction following the reviewers line of thought with the following changes:

We emphasize our main aim with an addition at L 50-52. “The aim of this work was to get a more fundamental understanding of the diversity, structure, metabolic potential and dynamics of the anaerobic benzene-degrading microbial community. **More specifically, we aimed at getting insight in the driving forces behind the persistence of large biodiversity in natural environments.**”

We add the following text to be more precise on the objectives of the succession experiment,

L 57. “An additional experiment was designed to independently identify the main organisms that drive anaerobic benzene degradation **as well to explore the metabolism of the culture.**”

We updated the overview of our findings and emphasised what is a result and what is the hypothesis, L 60-62.

“As such we i) identified the drivers for benzene degradation, ii) **got insight in the niches of the community members in the bioreactor and iii) hypothesized about the relevance of niche partitioning and microbial interactions** in order to explain the unexpected diversity of species in a bioreactor fed with benzene as main source of carbon and free energy.”

We emphasized an example that suggested niche partitioning with the following addition at discussion L 192-193. “**Additionally, Fig. 1b and Fig. 2c shows that the two primary consumers of benzene are significantly different from each other in their overall potential functionality. We, therefore, suggest niche partitioning for these two benzene degrading organisms.**”

Title:

The title needs to be more specific.

The title was:

Biodiversity and niche partitioning in an anaerobic benzene degrading culture

We changed it into a more specific one:

“High biodiversity is sustained by a few primary consumers in a benzene-degrading nitrate-reducing culture”

Abstract

The abstract needs to be more specific. Please explain the evidence that niche partitioning was uncovered from their data and that niche partitioning maintained high diversity and complexity. Specifically, what analyses were done and what results support the claims. As I explained above, I could not find any evidence or statistical support to the statements in Abstract from their Results.

We agree that there is no evidence that niche partitioning maintained high diversity and complexity. We rewrote the abstract and modified the discussion to highlight that niche partitioning is a hypothesis to explain biodiversity.

“A key question in microbial ecology is what the driving forces behind the persistence of large biodiversity in natural environments are. We studied a microbial community with more than 100 different types of species which evolved in a 15-years old bioreactor with benzene as the main carbon and free energy source and nitrate as the electron acceptor. We demonstrate by using genome-centric metagenomics plus metatranscriptomics that most of the community members most likely feed on metabolic left-overs or on necromass, while only a few of them are candidates to degrade benzene. We verify with an additional succession experiment using metabolomics and metabarcoding that these few community members are the actual drivers of benzene degradation. As such, we hypothesize that high species richness is maintained and the complexity of a natural community is stabilized in a controlled environment by the interdependencies between the few benzene degraders and the rest of the community members, ultimately resulting in a food web with different trophic levels.”

L20: "These mechanisms may well be conserved across ecosystems."

I am not sure what the authors imply here?

We understand the reviewer. That sentence is a hypothesis and since it builds on another hypothesis we skipped this sentence from the abstract.

Introduction

L53-55: "An additional experiment was designed to independently identify the main organisms that drive anaerobic benzene degradation. To this end, we inoculated a series of batch cultures of the 15 years-old microbial community at low cell densities." It's unclear why and how the authors test their hypothesis through a series of batch cultures. For example, how do species interact (such as competition for resources,

collaboration by exchange of products, inhibition in a chemical warfare and spatial organization) resulting in the compound patterns like Figure 3. There was no clear explanation in Methods how to quantify interactions and niche partitioning, either.

The succession experiment was designed to identify the main drivers of the benzene degrading community. Interactions between species and niche partitioning are hypotheses and not studied quantitatively. We changed the text to stress where we are conclusive, and where we hypothesize.

We changed L 29 because we believe it introduces confusion about on the objective of this study: “Here we approached this question by studying a” to “Here we study a”

Results

L70-73: "The majority of the MAGs was found to have a significantly"

This sentence regarding RNA abundance is uninformative. Moreover, I suspect that this result could be biased because the metagenomes (DNA as the template) were derived only from the biofilm, whereas the transcriptomes (RNA) were taken from both the biofilm and the liquid (under-represented in the template).

The reviewer is right to suspect a bias and it was investigated in supplementary Figure 1. This difference does not introduce bias in the MAG reconstructions and for that reason we excluded the DNA abundance from our analysis and only used it as a template. As a result, we only make statements about organisms that we observed and that were highly reconstructed based on the DNA and the mRNA. We have modified the phrase at L 76-77 to “32 out of the 47 MAGs were found to have a significantly ...”.

Before L128 vs. After L128:

The transition and linkage between “results based on metagenome data in the original bioreactor” vs. “results based on the 16S rDNA gene in newly generated batch grown cultures” are required for addressing the research objectives.

We modify L136-142 to address the transition between the two results and highlight the research objectives of the second:

“A succession experiment was performed to investigate the metabolism of the community as well as to identify the drivers of benzene degradation after giving them a fresh start. For that, we grew highly diluted batch cultures from the original bioreactor with benzene as carbon and free energy source and nitrate as electron acceptor. The cultures were sacrificed at different time intervals up to 34 days after inoculation and

analyzed for the community composition on the one hand and concentration of the metabolites on the other hand.”

The linkage between the two results is the finding that the member of the Peptococcaceae is not only a driver in the batch, but also the most active benzene degrader in the bioreactor. We have stated that more precisely in the text and we suggest the following changes:

Results

L 169-170 “Not surprisingly, the representative sequence of OTU624837510 from the batch grown cultures is found to be identical to the 16S rRNA gene of MAG 9 from the original bioreactor.”

Discussion

L 255-256 “A partial confirmation of this identity is that the 16S rRNA sequence of the most abundant OTU from the batch grown cultures was found to be identical to the one of the Peptococcaceae in MAG 9 from the original bioreactor.”

Discussion

L166-169: “We hypothesize that they interact with each other via syntrophy, scavenging, predation or even cheating (Fig. 5).....”

Here, in the discussion, the authors finally mention their “hypothesis”, with Fig. 5 to summarizing their findings in metagenome-assembled genomes (MAGs). Again, this is their hypothesis but not their findings.

We agree with the reviewer that this is our hypothesis. We addressed that in the previous points on the list.

We changed the caption of Figure 5 to reviewer point:

(b) Schematic representation of dominant niches and their hypothetical interactions.

L227-228: “The succession experiment and downstream correlation”

The authors provided succession data. However, instead providing new insights regarding species interactions, these data were mainly used to postulate the compound consumption in different stages and support the results of MAGs (primary consumers of benzene members were OTUs belonging to the family of Peptococcaceae (MAG 9) and Rhodocyclaceae (MAG 6) in the bioreactor). Thus again, the authors did not analyze interactions and niche partitioning.

Again, we agree with the reviewer. We did not analyze interactions and niche partitioning, these are our hypotheses. As stated earlier, we have stressed that in the new version of the manuscript.

L236-246: “Overall, our integrative systems ecology approach.....”

The same as the context in the Abstract, the whole story about “niche partitioning” and “a food web with different trophic levels” seems only according to the genome-centric hypothetical scheme (Fig. 5). This is a hypothesis but not their finding.

Again, we agree with the reviewer. We did not analyze interactions and niche partitioning, these are our hypotheses. As stated earlier, we have stressed that in the new version of the manuscript.

L 257-262 is modified to:

“Overall, our integrative systems ecology approach revealed that many different niches are occupied in the anoxic benzene degrading bioreactor. *We hypothesize that* niche partitioning in turn results in a community with a relatively high biodiversity (Fig. 5 a). Only a few species appear to metabolize benzene or breakdown products thereof. Yet, a wide range of microorganisms do not seem to feed on benzene but most likely on intermediates of benzene degradation, on biomolecules of *microbial necromass*, or autotrophically with ammonium as free energy source. As a result, most of the community members make up a specialized food web with different trophic levels despite the limited resources. “

We deleted the following phrase at L 262-264: “*We envisage similar organizations of and interactions within communities across many different ecosystems. Whether or not such communities are driven by key species as we have shown for the community in this study remains to be elucidated.*”

Reviewers' comments:

Reviewer #1 (Remarks to the Author):

Title: Improved, but it makes more sense that way:

High biodiversity in a benzene-degrading nitrate-reducing culture is sustained by a few primary consumers

Abstract Remove one of the most ... and maybe start your green part with "Using genomic... we," and in total three "we" sentences in a row is a bit much.

The abstract should have at least some concrete results – which organism are doing benzene degradation, what do others do... i.e. Rhodocyclaceae (MAG 6) and the Peptococcaceae (MAG 9) are likely the organisms involved in benzene degradation

I am still very critical about the high number of supplementary figures. I am not sure that they are all needed – and 37 Supp Figures, excessive text and tables that is truly excessive.

Especially if the result are rather moderate: You have a community based mainly on 2 benzene degraders, an anammox organism, and some potentially predatory/ biofilm producing bacteria organisms Chloroflexi does not seem to do benzene degradation (SFig. 28 or?). Further reasons for the different community composition of liquid and biofilms are not discussed.

If all the supplementary figures have important results, please state it. If these figures have a relevance, please explain the outcome in the legend

Line 182: you speak of MAGs – those do not interact via syntrophy... maybe they interact in your metagenomic analyses.. it's the organisms here.

Line 186: I don't think you can compare the transcription in the biofilm with the liquid phase in this way ... What I see in the data is that the biofilm has higher diversity, whereas the liquid phase has a single dominant organism that is presumably the only one that is growing fast enough not to be diluted out (doubles faster than every 100 hours (1/ dilution rate) under given conditions in the reactor. In the biofilm a far more complex community can establish. The film assimilates all organics that are released into the biofilm..

Fig. 3 ...sampled at different time points.

Reviewer #2 (Remarks to the Author):

See attachment

After checking the new revised version of the paper I believe that the Authors have made a huge effort in order to improve the manuscript. The last version includes modifications that answer to the criticisms previously raised by the reviewers. I do not have other additional comments to add and I feel the manuscript in the present form is suitable for publication.

Sincerely

We would like to thank the reviewers for reading and assessing our work once more. The text with blue color indicates the reviewers points and the text in black color our responses. All changes (such as addition, deletion, or replacements) below as well in the updated manuscript are indicated with green color.

Referee expertise:

Referee #1: microbial hydrocarbon metabolism, biochemistry and molecular ecology

Referee #2: broad interests in genomics and metagenomics.

Reviewers' comments:

Reviewer #1 (Remarks to the Author):

Title: Improved, but it makes more sense that way:

High biodiversity in a benzene-degrading nitrate-reducing culture is sustained by a few primary consumers

We found the suggested change an improvement and we included it.

Abstract Remove one of the most ... and maybe start your green part with "Using genomic... we," and in total three "we" sentences in a row is a bit much.

We agree and followed the suggestion.

The abstract should have at least some concrete results – which organism are doing benzene degradation, what do others do... i.e. Rhodocyclaceae (MAG 6) and the Peptococcaceae (MAG 9) are likely the organisms involved in benzene degradation

We agree and followed the suggestion.

I am still very critical about the high number of supplementary figures. I am not sure that they are all needed – and 37 Supp Figures, excessive text and tables that is truly excessive.

We have had a critical look at the supplements. We skipped Table S4 and S9, Figures S3, S8, and S33 (old enumeration).

Most of the others are analyses in-depth about the occurrence and metabolic properties of the community members and give us and the readers a more fundamental understanding of the

community. Also, it addresses the next point of the reviewer, where supplementary Fig. S26 (updated enumeration) provides the arguments for the role of Chloroflexi (see below). Most of the supplementary text also gives an in-depth insight into pathways of benzene degradation and nitrogen cycling reactions in the culture, both key to central metabolism in the community.

The supplementary material is crucial for issues of reproducibility, clearance, and support of the hypothesis stated in the main article.

Especially if the result are rather moderate: You have a community based mainly on 2 benzene degraders, an anammox organism, and some potentially predatory/ biofilm producing bacteria organisms Chloroflexi does not seem to do benzene degradation (SFig. 28 or?). Further reasons for the different community composition of liquid and biofilms are not discussed.

Following the observation of the reviewer in Fig. S26 (updated enumeration) the reader can see the low transcription nodes in the graph of Chloroflexi (MAG 01). The two nodes namely, "Benzoyl CoA syn 01" and "Mid aerobic" are presented in more detail in Fig. S12 and S16 (updated enumeration) where the reader can look up the corresponding functions, for example, on Chloroflexi (MAG 01)

Those are:

R05587 Benzoylsuccinyl-CoA + CoA <=> Benzoyl-CoA + Succinyl-CoA

K07549 benzoylsuccinyl-CoA thiolase BbsA subunit

R00829 Succinyl-CoA + Acetyl-CoA <=> CoA + 3-Oxoadipyl-CoA

K00632 acetyl-CoA acyltransferase

K02615 3-oxo-5,6-didehydrosuberyl-CoA/3-oxoadipyl-CoA thiolase

Also, we reported at results L 104-105:

"Notably, all MAGs from the Chloroflexi expressed relatively high levels of mRNAs encoding peptidases, extracellular solute-binding proteins and specific ABC-type transporters."

Together those results lead us to the hypothesis that Chloroflexi behaves as a scavenger of metabolic left-overs (perhaps benzoylsuccinyl-CoA) or it feeds on necromass.

We would also like to mention our discussion about the different community composition of liquid and biofilm at L 176-178: "Maintenance of and interactions within this relatively complex microbial community may be envisaged in this type of bioreactor where a continuous flow system is in contact with the biofilm. As such, it may host species that grow at much lower rates than the dilution rate [9, 39]."

To discuss the implication of the sustained diversity in the bioreactor over such long timescales we added the following text in the discussion L 179-185:

Also, the biofilm may have a high potential for spatial heterogeneity, further contributing to a higher biodiversity (Walters et al 2020, Stalder et al 2020). In addition, it may confer functional stability since it can shield important functional groups from disturbances and as such may be less prone to invasion by taxa that could interfere with community functioning (Battin et al, 2007, 2016, Gralka et al, 2020). The high biodiversity sustained in the bioreactor over such long timescales ensures metabolic plasticity of the microbial community, which is important for adaptive responses to environmental changes such as the supply of electron donors, electron acceptors, or nutrients. Indeed, the same consortium was shown to readily use iron(III) or sulfate instead of nitrate as the electron acceptor for benzene degradation (Van der Zaan et al 2012).

If all the supplementary figures have important results, please state it. If these figures have a relevance, please explain the outcome in the legend

For most supplementary Figures, we added the outcome as requested by the reviewer

Line 182: you speak of MAGs – those do not interact via syntrophy... maybe they interact in your metagenomic analyses.. it's the organisms here.

We agree and suggest the following phrasing at L 175-176:

We hypothesize that the organisms represented by those MAGs interact with each other via syntrophy...

Line 186: I don't think you can compare the transcription in the biofilm with the liquid phase in this way ... What I see in the data is that the biofilm has higher diversity, whereas the liquid phase has a single dominant organism that is presumably the only one that is growing fast enough not to be diluted out (doubles faster than every 100 hours (1/ dilution rate) under given conditions in the reactor. In the biofilm a far more complex community can establish. The film assimilates all organics that are released into the biofilm..

We can compare them as we normalized the data accordingly (Methods Multi-omics analysis, equation (4)). Apart from MAG 9, we found another dominant organism in the liquid phase, namely MAG 18, and two less dominant MAGs 30 and 69. For the rest, we fully agree with the reviewer as discussed in our original manuscript.

Fig. 3 ...sampled at different time points.

We followed the suggestion.

Reviewer #2 (Remarks to the Author):

After checking the new revised version of the paper I believe that the Authors have made a huge effort in order to improve the manuscript. The last version includes modifications that answer to the criticisms previously raised by the reviewers. I do not have other additional comments to add and I feel the manuscript in the present form is suitable for publication.

Sincerely

We thank the reviewer.